

# Nitrogen isotopic fractionations during nitric oxide production in an agricultural soil

Zhongjie Yu[1,2] and Emily M. Elliott[1]

[1]Department of Geology and Environmental Science, University of Pittsburgh, Pittsburgh, Pennsylvania 15260, USA

[2]Department of Natural Resources and Environmental Sciences, University of Illinois Urbana-Champaign, Urbana, Illinois 61801, USA.

*Correspondence to*: Zhongjie Yu (zjyu@illinois.edu)

**Abstract.** Nitric oxide (NO) emissions from agricultural soils play a critical role in atmospheric chemistry and represent an important pathway for loss of reactive nitrogen (N) to the environment. With recent methodological advances, there is growing interest in the natural abundance N isotopic composition ($\delta^{15}N$) of soil-emitted NO and its utility in providing mechanistic information on soil NO dynamics. However, interpretation of soil $\delta^{15}N$-NO measurements has been impeded by the lack of constraints on the isotopic fractionations associated with NO production and consumption in relevant microbial and chemical reactions. In this study, anoxic (0% $O_2$), oxic (20% $O_2$), and hypoxic (0.5% $O_2$) incubations of an agricultural soil were conducted to quantify the net N isotope effects ($^{15}\eta$) for NO production in denitrification, nitrification, and abiotic reactions of nitrite ($NO_2^-$) using a newly developed $\delta^{15}N$-NO analysis method. A sodium nitrate ($NO_3^-$) containing mass-independent oxygen-17 excess (quantified by a $\Delta^{17}O$ notation) and three ammonium ($NH_4^+$) fertilizers spanning a $\delta^{15}N$ gradient were used in soil incubations to help illuminate the reaction complexity underlying NO yields and $\delta^{15}N$ dynamics in a heterogeneous soil environment. We found strong evidence for the prominent role of $NO_2^-$ oxidation under anoxic conditions in controlling the apparent $^{15}\eta$ for NO production from $NO_3^-$ in denitrification (i.e., 49 to 60‰). These results highlight the importance of an under-recognized mechanism for the reversible enzyme $NO_2^-$ oxidoreductase to control the N isotope distribution between the denitrification products. Through a $\Delta^{17}O$-based modeling of co-occurring denitrification and $NO_2^-$ re-oxidation, the $^{15}\eta$ for $NO_2^-$ reduction to NO and NO reduction to nitrous oxide ($N_2O$) were constrained to be 15 to 22‰ and -8 to 2‰, respectively. Production of NO in the oxic and hypoxic incubations was contributed by both $NH_4^+$ oxidation and $NO_3^-$ consumption, with both processes having a significantly higher NO yield under $O_2$ stress. Under both oxic and hypoxic conditions, NO production from $NH_4^+$ oxidation proceeded with a large $^{15}\eta$ (i.e., 55 to 84‰) possibly due to expression of multiple enzyme-level isotopic fractionations during $NH_4^+$ oxidation to $NO_2^-$ that involves NO as either a metabolic byproduct or an obligatory intermediate for $NO_2^-$ production. Adding $NO_2^-$ to sterilized soil triggered substantial NO production, with a relatively small $^{15}\eta$ (19‰). Applying the estimated $^{15}\eta$ values to a previous $\delta^{15}N$ measurement of in situ soil $NO_x$ emission ($NO_x = NO+NO_2$) provided promising evidence for the potential of $\delta^{15}N$-NO measurements in revealing NO production pathways. Based on the observational and modeling constraints obtained in this study, we suggest that simultaneous $\delta^{15}N$-NO and $\delta^{15}N$-$N_2O$ measurements can lead to unprecedented insights into the sources of and processes controlling NO and $N_2O$ emissions from agricultural soils.



## 1 Introduction

Agricultural production of food has required a tremendous increase in the application of nitrogen (N) fertilizers since 1960s (Davidson, 2009). In order to maximize crop yields, N fertilizers are often applied in excess to agricultural soils, resulting in loss of reactive N to the environment (Galloway et al., 2003). Loss of N in the form of gaseous nitric oxide (NO) has long been recognized for its adverse impacts on air quality and human health (Veldkamp and Keller, 1997). Once emitted to the atmosphere, NO is rapidly oxidized to nitrogen dioxide ($NO_2$), and these compounds (collectively referred to $NO_x$) drive production and deposition of atmospheric nitrate ($NO_3^-$) (Calvert et al., 1985) and play a critical role in the formation of tropospheric ozone ($O_3$) – a toxic air pollutant and potent greenhouse gas (Crutzen, 1979). Despite the observations that emission of NO from agricultural soils can sometimes exceed that of nitrous oxide ($N_2O$) – a climatically important trace gas primarily produced from reduction of NO in soils (Liu et al., 2017), NO is frequently overlooked in soil N studies due to its high reactivity and transient presence relative to $N_2O$ (Medinets et al., 2015). Consequently, the contribution of soil NO emission to contemporary $NO_x$ inventories at regional to global scales is highly uncertain (e.g., ranging from 3% to >30%) (Hudman et al., 2010; Vinken et al., 2014) and remains the subject of much current debate (Almaraz et al., 2018; Maaz et al., 2018).

As the "central hub" of the biogeochemical N cycle, NO can be produced and consumed in numerous microbial and chemical reactions in soils (Medinets et al., 2015). Among these processes, nitrification and denitrification are the primary sources responsible for NO emission from N-enriched agricultural soils (Firestone and Davidson, 1989). Denitrification is the sequential reduction of $NO_3^-$ and nitrite ($NO_2^-$) to NO, $N_2O$, and dinitrogen ($N_2$) and can be mediated by a diversity of soil heterotrophic microorganisms (Zumft, 1997). The enzymatic system of denitrification comprises a series of dedicated reductases whereby $NO_2^-$ reductase (NIR) and NO reductase (NOR) are the key enzymes that catalyze production and reduction of NO, respectively (Ye et al., 1994). As such, NO is often viewed as a free intermediate of the denitrification process (Russow et al., 2009). In comparison, nitrification is a two-step aerobic process, in which oxidation of ammonia ($NH_3$) to $NO_2^-$ is mediated by ammonia-oxidizing bacteria (AOB) or archaea (AOA), while the subsequent oxidation of $NO_2^-$ to $NO_3^-$ is performed by nitrite-oxidizing bacteria (NOB) (Lehnert et al., 2018). Although production of NO during the nitrification process has been linked to $NH_3$ oxidation (Hooper et al., 2005; Caranto et al., 2017) and $NO_2^-$ reduction by AOB/AOA-encoded NIR (Wrage-Mönning et al., 2018), the metabolic role of NO in AOB and AOA remains ambiguous, making it difficult to elucidate the enzymatic pathways driving NO release by nitrification (Beeckman et al., 2018; Stein, 2019). Additionally, NO can also be produced from abiotic reactions involving soil $NO_2^-$ or its protonated form – nitrous acid ($HNO_2$) (Venterea et al., 2005; Lim et al., 2018). However, despite empirical evidence for the dependence of soil NO emission on soil N availability and moisture content (Davidson and Verchot, 2000), the source contribution of soil NO emission across temporal and spatial scales is poorly understood (Hudman et al., 2012). This is largely due to the lack of a robust means for source partitioning soil-emitted NO under dynamic environmental conditions.

Natural abundance stable N and oxygen (O) isotopes in N-containing molecules have long provided insights into the sources and relative rates of biogeochemical processes comprising the N cycle (Granger and





Wankel, 2016). The unique power of stable isotope ratio measurements stems from the distinct partitioning of isotopes between chemical species or phases, known as isotopic fractionation. Thus, in order to extract the greatest

information from the distributions of isotopic species, a rigorous understanding of the direction and magnitude of isotopic fractionations associated with each relevant transformation is required. Both kinetic and equilibrium isotope effects can lead to isotopic fractionations between N-bearing compounds in soils (Granger and Wankel, 2016; Denk et al., 2017). During kinetic processes, isotopic fractionation occurs as a result of differences in the reaction rates of isotopically substituted molecules, leading to either enrichment or, in a few rare cases, depletion of heavy isotopes in

the reaction substrate (Fry, 2006; Casciotti, 2009). The degree of kinetic fractionation can be quantified by a kinetic fractionation factor ($\alpha_k$), which is often represented by the ratio of reaction rate constant of light isotopes to that of heavy isotopes. In this definition, $\alpha_k$ is larger than 1 for a normal kinetic fractionation. For equilibrium reactions, equilibrium fractionation arises from differences in the zero-point energies of two species undergoing isotopic exchange, leading to enrichment of heavy isotopes in the more strongly bonded form (Fry, 2006; Casciotti, 2009). In

this case, the isotope ratios of two species at equilibrium are defined by an equilibrium fractionation factor ($\alpha_{eq}$), which is also related to the kinetic fractionation factors of forward and backward equilibrium reactions (Fry, 2006). By convention, isotopic fractionation can be expressed in units of per mille (‰) as an isotope effect ($\varepsilon$): $\varepsilon = (\alpha - 1) \times 1000$. Nevertheless, in a heterogeneous soil environment, expression of intrinsic kinetic and equilibrium isotope effects for biogeochemical N transformations is often limited due to transport limitation in soil substrates, the multi-

step nature of transformation processes, as well as presence of diverse soil microbial communities that transform N via parallel and/or competing reaction pathways (Maggi and Riley, 2010). As such, interpretation of N isotope distribution in soils has largely relied on measuring net isotope effects ($\eta$), which are often characterized by incubating soil samples under environmentally relevant conditions, that favor expression of intrinsic isotope effects for specific N transformations (Lewicka-Szczebak et al. 2014). For example, it has been shown that the net N

isotope effects for $N_2O$ production in soil nitrification, denitrification, and abiotic reactions are distinctively different under certain soil conditions (Denk et al., 2017), rendering natural abundance N isotopes of $N_2O$ a useful index for inferring sources of $N_2O$ in agricultural soils (Toyoda et al., 2017).

      While the isotopic dynamics underlying soil $N_2O$ emissions has been extensively studied, there has been little investigation into the N isotopic composition (notated as $\delta^{15}N$ in units of ‰; $\delta = ((R_{sample}/R_{standard})-1)\times1000$) of

soil-emitted NO due to measurement difficulties (Yu and Elliott, 2017). Using a tubular denuder that trapped NO released from urea and ammonium ($NH_4^+$)-fertilized soils, Li and Wang (2008) revealed a gradual increase in $\delta^{15}N$-NO from -49 to -19‰ and simultaneous $^{15}N$ enrichment in soil $NH_4^+$ and $NO_3^-$ over a two-week laboratory incubation. Similar $\delta^{15}N$ variations (i.e., -44 to -14‰) were recently reported for in situ soil $NO_x$ emission in a manure-fertilized cornfield (Miller et al., 2018). Moreover, the magnitude of $\delta^{15}N$-$NO_x$ measured in this study

depended on manure application methods, implying that $NO_x$ was mainly sourced from nitrification of manure-derived $NH_4^+$ (Miller et al., 2018). Based on a newly developed soil NO collection system that quantitatively converts soil-emitted NO to $NO_2$ for collection in triethanolamine (TEA) solutions, our previous work demonstrated substantial variations in $\delta^{15}N$-NO (-54 to -37‰) in connection with changes in moisture content in a forest soil (Yu and Elliott, 2017). Furthermore, the measured in situ $\delta^{15}N$-NO values spanned a wide range (-60 to -23‰) and were





highly sensitive to added N substrates (i.e., $NH_4^+$, $NO_3^-$, and $NO_2^-$), indicating that NO produced from different
       sources may bear distinguishable $\delta^{15}$N imprints (Yu and Elliott, 2017). Nevertheless, despite the potential of $\delta^{15}$N-
       NO measurements in providing mechanistic information on soil NO dynamics, interpretation of $\delta^{15}$N-NO has been
       largely impeded by the knowledge gap as to how $\delta^{15}$N-NO is controlled by N isotopic fractionations during NO
       production and consumption in soils.

To this end, we conducted a series of controlled incubation experiments to quantify the net N isotope
       effects for NO production in an agricultural soil. Replicate soil incubations were conducted to measure the yield and
       $\delta^{15}$N of soil-emitted NO under anoxic (0% $O_2$), oxic (20% $O_2$), and hypoxic (0.5% $O_2$) conditions, respectively. A
       sodium $NO_3^-$ fertilizer mined in the Atacama Desert, Chile (Yu and Elliott, 2018) was used to amend the soil in all
       three incubation experiments. This Chilean $NO_3^-$ originated from atmospheric deposition and thus contained an
anomalous $^{17}$O excess (quantified by a $\Delta^{17}$O notation) as a result of mass-independent isotopic fractionations during
       its photochemical formation in the atmosphere (Michalski et al., 2004). Because isotopic fractionations during
       biogeochemical $NO_3^-$ production and consumption are mass-dependent, $\Delta^{17}$O-$NO_3^-$ is a conservative tracer of gross
       nitrification and $NO_3^-$ consumption and provides a quantitative benchmark for disentangling isotopic overprinting on
       $\delta^{15}$N-$NO_3^-$ and $\delta^{18}$O-$NO_3^-$ during co-occurring nitrification and denitrification (Yu and Elliott, 2018) (see Text S1 in
the Supplement for more details). As additional tracers, three isotopically different $NH_4^+$ fertilizers were used in
       parallel treatments of the oxic and hypoxic incubations to quantify the source contribution of NO production with
       changing $O_2$ availability. By integrating multi-species measurements of N and O isotopes in an isotopologue-
       specific modeling framework, we were able for the first time to unambiguously link the yield and $\delta^{15}$N variations of
       soil-emitted NO to nitrification and denitrification carried out by whole soil microbial communities and to
characterize the net isotope effects for NO production from soil $NO_3^-$, $NH_4^+$, and $NO_2^-$ under different redox
       conditions. The quantified isotope effects are discussed in the context of chemical and enzymatic pathways leading
       to net NO production in the soil environment and are applied to a previous field study (Miller et al., 2018) to provide
       implications for tracing the sources of NO emission from agricultural soils.

## 2 Materials and methods

### 2.1 Soil characteristics and preparation

       Soil samples used in this study were collected in July 2017 from a conventional corn-soybean rotation field in
       central Pennsylvania, USA managed by the USDA (Agricultural Research Service, University Park, PA, USA). The
       soil is a well-drained Hagerstown silt loam (fine, mixed, semiactive, mesic Typic Hapludalfs) with sand, silt, and
       clay content of 21%, 58%, and 21%, respectively. The sampled surface layer (0 - 10 cm) had a bulk density of 1.2
g·$cm^{-3}$ and a pH (1:1 water) of 5.7. Total N content was 0.2% and $\delta^{15}$N of total N was 5.3‰. Soil C:N ratio was 11.4
       and organic carbon content was 1.8%. In the laboratory, soils were homogenized and sieved to 2 mm (but not air-
       dried) and then stored in resealable plastic bags at 4˚C until further analyses and incubations. Gravimetric water
       content of the sieved and homogenized soils was 0.14 g $H_2O$·$g^{-1}$. Indigenous $NH_4^+$ and $NO_3^-$ concentrations were 0.7





μg N·g$^{-1}$ and 19.8 μg N·g$^{-1}$, respectively. Throughout this paper, soil N concentrations, NO fluxes, and N
transformation rates are expressed on the basis of soil oven-dry (105°C) weight.

## 2.2 Net NO production and collection of NO for δ$^{15}$N analysis

The recently developed soil dynamic flux chamber (DFC) system was used to measure net NO production rates and
to collect soil-emitted NO for δ$^{15}$N analysis (Yu and Elliott, 2017). A schematic of the DFC system is shown in Fig.
1a. Detailed development and validation procedures for the NO collection method were presented in Yu and Elliott
(2017). Briefly, custom-made flow-through incubators modified from 1 L Pyrex medium bottles (13951L, Corning,
USA) were used for all the incubation experiments (Fig. 1b). Each incubator was stoppered with two 42 mm Teflon
septa secured by an open-topped screw cap and equipped with two vacuum valves for purging and closure of the
incubator headspace. To measure net NO production from enclosed soil samples, a flow of NO-free air with desired
$O_2$ content was directed through the incubator into a chemiluminescent NO-NO$_x$-NH$_3$ analyzer (model 146i, Thermo
Fisher Scientific) (Fig. 1a) (Yu and Elliott, 2017). Outflow NO concentration was monitored continuously until
steady and then the net NO production rate was determined from the flow rate and steady-state NO concentration.
To collect NO for δ$^{15}$N analysis, a subsample of the incubator outflow was forced to pass through a NO collection
train (Fig. 1a) where NO is converted to $NO_2$ by excess $O_3$ (~3 ppm) in a Teflon reaction tube (9.5 mm I.D., ca. 240
cm length) and subsequently collected in a 500 mL gas washing bottle containing a 20% (v/v, 70 mL) TEA solution
(Yu and Elliott, 2017). The collection products were about 90% $NO_2^-$ and 10% $NO_3^-$ (Yu and Elliott, 2017). Results
from comprehensive method testing showed that the NO collection efficiency was 98.5±3.5% over a wide range of
NO concentrations (12 to 749 ppb) and environmental conditions (e.g., temperature from 11 to 31°C and relative
humidity of the incubator outflow from 27 to 92%) (Yu and Elliott, 2017). Moreover, it was confirmed that high
concentrations of ammonia ($NH_3$) (e.g., 500 ppb) and nitrous acid (HONO) (removed by an inline HONO scrubber
(Fig. 1a)) in the incubator outflow do not interfere with NO collection (Yu and Elliott, 2017).

## 2.3 Anoxic incubation

To prepare for the anoxic incubation, the soil samples were spread out on a covered tray for pre-conditioning under
room temperature (21 °C) for 24 h. Next, the soil was amended with the Chilean $NO_3^-$ fertilizer (δ$^{15}$N=0.3±0.1‰,
δ$^{18}$O=55.8±0.1‰, Δ$^{17}$O=18.6±0.1‰) to achieve a fertilization rate of 35 μg $NO_3^-$-N·g$^{-1}$ and a target soil water
content of 0.21 g $H_2O$·g$^{-1}$ (equivalent to 46% water-filled pore space (WFPS)). The fertilized soil samples were
thoroughly homogenized using a glass rod in the tray. 100 g (dry weight equivalent) soil was then weighed into each
of eight incubators, resulting in a soil depth of about 1.5 cm. The incubators were connected in parallel using a
Teflon purging manifold (Fig. 1c), vacuumed and filled with ultra-purity $N_2$ for three cycles, and incubated in dark
with a continuous flow of $N_2$ circulating through each of the eight incubators at 0.015 standard liter per minute
(SLPM). The sample fertilization and preparation procedures were repeated three times to establish three batches of
replicate samples, leading to 24 soil samples in total for the anoxic incubation.

The first NO measurement and collection event was conducted 24 h after the onset of the anoxic incubation
and daily sampling was conducted thereafter. At each sampling event, one incubator from each replicate sample



batch was isolated by closing the vacuum valves, removed from the purging manifold, and then measured using the DFC system. To prevent $O_2$ contamination by residual air in the DFC system, the DFC system was evacuated and flushed with $N_2$ five times before the vacuum valves were re-opened. A flow of $N_2$ was then supplied at 1 SLPM for continuous NO concentration measurement and collection. Samples from the replicate batches were measured successively.

Following the completion of measurement and collection of each sample, the incubator was opened from the top and the soil was combined with 500 mL deionized water for extraction of soil $NO_3^-$ and $NO_2^-$ (McKenney et al., 1982). Because $NO_2^-$ accumulation was found in pilot experiments, deionized water, rather than routinely used KCl solutions, was used for the extraction to ensure accurate $NO_2^-$ determination (Homyak et al., 2015). To extract soil $NO_3^-$ and $NO_2^-$, the soil slurry was agitated vigorously on a stir plate for 10 minutes and then centrifuged for 10 minutes at 2000 rpm. The resultant supernatant was filtered through a sterile 0.2 µm filter (Homyak et al., 2015). In light of high $NO_2^-$ concentrations observed in the pilot experiments, the filtrate was divided into two 60 mL Nalgene bottles, with one of the bottles receiving sulfamic acid to remove $NO_2^-$ (Granger et al., 2009). This $NO_2^-$-removed sample was used for $NO_3^-$ isotope analysis, while the other sample without sulfamic acid treatment was used for determining $NO_2^-$ and $NO_3^-$ concentrations and combined $\delta^{15}N$ analysis of $NO_2^-+NO_3^-$. Two important control tests, based on $NO_2^-/NO_3^-$ spiking and acetylene ($C_2H_2$) addition, were conducted to evaluate the robustness of the adopted soil incubation and extraction methods. The results confirmed that the water extraction method was robust for determining concentrations and isotopic composition of soil $NO_3^-$ and $NO_2^-$ and that aerobic $NO_3^-$ production from $NH_4^+$ oxidation was negligible during the soil incubation and extraction procedures (Table S1 and Table S2; see Text S2 in the Supplement for more details).

### 2.4 Oxic and hypoxic incubations

The same pre-conditioning and fertilization protocol described for the anoxic incubation was used for the oxic and hypoxic incubations. Three isotopically different $NH_4^+$ fertilizers were used in parallel treatments of each incubation experiment: (1) $\delta^{15}N$-$NH_4^+$=1.9‰ (low $^{15}N$ enrichment), (2) $\delta^{15}N$-$NH_4^+$=22.5‰ (intermediate $^{15}N$ enrichment), and (3) $\delta^{15}N$-$NH_4^+$=45.0‰ (high $^{15}N$ enrichment). An off-the-shelf ammonium sulfate (($NH_4)_2SO_4$) reagent was used in the low $\delta^{15}N$-$NH_4^+$ treatment, while the fertilizers with intermediate and high enrichment of $^{15}N$ were prepared by gravimetrically mixing the ($NH_4)_2SO_4$ reagent with $NH_4^+$ reference materials IAEA-N2 ($\delta^{15}N$-$NH_4^+$=20.3‰) and USGS26 ($\delta^{15}N$-$NH_4^+$=53.7‰). In both oxic and hypoxic incubations, each of the three $\delta^{15}N$-$NH_4^+$ treatments consisted of three replicate sample batches where each batch consisted of eight samples, resulting in 72 samples for each incubation experiment.

At the onset of each incubation experiment, soil samples (100 g dry weight equivalent) were amended with desired $NH_4^+$ fertilizer (90 µg N·g$^{-1}$) and the Chilean $NO_3^-$ fertilizer (15 µg N·g$^{-1}$) to the target soil water content of 0.21 g $H_2O$·g$^{-1}$ (46% WFPS). Following the amendment, two soil samples from each replicate batch were immediately extracted – one with 500 mL of deionized water for soil $NO_2^-$ and $NO_3^-$ using the extraction method described above and the other one with 500 mL of a 2 M KCl solution for determination of soil $NH_4^+$. The remaining samples were incubated under desired $O_2$ conditions until further measurements. In the oxic incubation,



the incubators were connected in parallel using the purging manifold and continuously flushed by a flow of zero air (20% $O_2$ + 80% $N_2$). In the hypoxic incubation, a flow of synthetic air with 0.5% $O_2$ content (balanced by 99.5% $N_2$) was used to incubate the soil samples. The synthetic air was generated by mixing the zero air with ultra-purity $N_2$ using two mass flow controllers (Model SmartTrak 50, Sierra Instruments).

Replicate NO measurement and collection events were conducted at 24 h, 48 h, and 72 h following the onset of the oxic and hypoxic incubations. Because net NO production rates were low under oxic and hypoxic conditions, all remaining soil samples in each replicate batch were connected in parallel for NO measurement and collection using the DFC system. This parallel connection ensured high outflow NO concentrations (i.e., >30 ppb) required for quantitative NO collection (Yu and Elliott, 2017). The flow rate of purging air (20% $O_2$ for the oxic incubation and 0.5% $O_2$ for the hypoxic incubation) during the DFC measurement was 0.25 SLPM to each incubator. Following the NO measurement and collection, two soil samples from each replicate batch were extracted for determination of soil $NO_3^-/NO_2^-$ (500 mL deionized water) and $NH_4^+$ (500 mL 2M KCl), respectively. Because NO concentrations were too low for reliable NO collection at 72 h after the onset of the incubations, only net NO production rates were measured using the remaining two soil samples in each replicate batch.

**2.5 Abiotic NO production**

The potential for NO production from abiotic reactions was assessed using sterilized soil samples. Soil samples (100 g dry-weight equivalent) were weighed into the incubators and then autoclaved at 121˚C and 1.3 atm for 30 minutes. The autoclaved samples were pre-incubated under oxic and anoxic conditions, respectively, for 24 h and then fertilized with the Chilean $NO_3^-$ (35 µg $NO_3^-$-N·g$^{-1}$) or the lab $(NH_4)_2SO_4$ (90 µg $NH_4^+$-N·g$^{-1}$). The fertilizer solutions were added to the soil surface through the Teflon septa using a sterile syringe equipped with a 25-gauge needle. These samples were then measured periodically for net NO production. Because $NO_2^-$ was found to accumulate during the anoxic incubation (see below), four soil samples were sterilized, pre-incubated under anoxic condition, and then fertilized with a $NaNO_2$ solution ($\delta^{15}N$-$NO_2^-$=1.4±0.2‰) (8 µg N·g$^{-1}$) for immediate NO measurement and collection. These $NO_2^-$-amended samples were thereafter incubated under anoxic conditions and measured periodically for net NO production until undetectable.

**2.6 Chemical and isotopic analyses**

Soil $NO_3^-$ concentrations were determined using a Dionex Ion Chromatograph ICS-2000 with a precision of (1σ) of ±5.0 µg N·L$^{-1}$. Soil $NO_2^-$ concentrations were analyzed using the Greiss-Islovay colorimetric reaction with a precision of ±1.2 µg N·L$^{-1}$. Soil $NH_4^+$ concentrations were measured using a modified fluorometric OPA method for soil KCl extracts (Kang et al., 2003) with a precision of ±7.0 µg N·L$^{-1}$. $NO_2^-$+$NO_3^-$ concentration in the TEA collection samples was measured using a modified spongy cadmium method with a precision of ±1.6 µg N·L$^{-1}$ (Yu and Elliott, 2017).

The denitrifier method (Sigman et al., 2001; Casciotti et al., 2002) was used to measure $\delta^{15}N$ and $\delta^{18}O$ of $NO_3^-$ in the $NO_2^-$-removed soil extracts and the $\delta^{15}N$ of $NO_3^-$+$NO_2^-$ in the extracts without sulfamic acid treatment. In brief, a denitrifying bacterium (*Pseudomonas aureofaciens*) lacking the $N_2O$ reductase enzyme was used to





convert 20 nmol of $NO_3^-$ into gaseous $N_2O$. The $N_2O$ was then purified in a series of chemical traps, cryo-focused, and finally analyzed on a GV Instruments Isoprime Continuous Flow Isotope Ratio Mass Spectrometer (CF-IRMS) at *m/z* 44, 45, and 46 at the University of Pittsburgh *Regional Stable Isotope Laboratory for Earth and Environmental Science Research* where all isotope analyses were conducted for this study. International $NO_3^-$ reference standards IAEA-N3, USGS34, and USGS35 were used to calibrate the $\delta^{15}N$ and $\delta^{18}O$ analyses. The long-

term precision is ±0.3‰ and ±0.5‰, respectively, for the $\delta^{15}N$ and $\delta^{18}O$ analyses. Because the denitrifier method does not differentiate $NO_3^-$ and $NO_2^-$ for the $\delta^{15}N$ analysis, $\delta^{15}N$ of $NO_2^-$ was estimated using an isotopic mass balance when $NO_2^-$ accounted for a significant fraction of the total $NO_3^-$+$NO_2^-$ pool.

        $\Delta^{17}O$ of $NO_3^-$ was measured using the coupled bacterial reduction and thermal decomposition method described by Kaiser et al. (2007). The denitrifying bacteria were used to convert 200 nmol of $NO_3^-$ to $N_2O$, which

was subsequently converted to $O_2$ and $N_2$ by reduction over a gold surface at 800 °C. The produced $O_2$ and $N_2$ were separated using a 5Å molecular sieve gas chromatograph, and the $O_2$ was then analyzed for $\delta^{17}O$ and $\delta^{18}O$ using the CF-IRMS. $\Delta^{17}O$ was calculated from the measured $\delta^{17}O$ and $\delta^{18}O$ using Equation (1) (see Text S1 in the Supplement) and calibrated by USGS34, USGS35, and a 1:1 mixture of USGS34 and USGS35.

$$\Delta^{17}O = \left[ \ln\left(\frac{\delta^{17}O}{1000} + 1\right) - 0.52 \ln\left(\frac{\delta^{18}O}{1000} + 1\right) \right] \times 1000 \qquad \text{Equation (1)}$$

The precision of the $\Delta^{17}O$ analysis of USGS35 and the USGS35:USGS34 mixture is ±0.3‰ (Yu and Elliott, 2018). Following Kaiser et al. (2007), the measured $\Delta^{17}O$-$NO_3^-$ was used in the reduction of molecular isotope ratios of $N_2O$ to correct for the isobaric interference (i.e., *m/z* 45) on the measured $\delta^{15}N$-$NO_3^-$.

        $\delta^{15}N$ of $NH_4^+$ in the KCl extracts was measured by coupling the $NH_3$ diffusion method (Zhang et al., 2015) and the hypobromite ($BrO^-$) oxidation method (Zhang et al., 2007) with the denitrifier method (Felix et al., 2013).

Briefly, an aliquot of soil KCl extract with 60 nmol $NH_4^+$ was pipetted into a 20 mL serum vial containing an acidified glass fiber disk. The solution was made alkaline by adding magnesium oxide (MgO) to volatilize $NH_3$, which was subsequently captured on the acidic disk as $NH_4^+$. After incubation under 37 °C for 10 d, $NH_4^+$ was eluted from the disk using deionized water, diluted to 10 µM, oxidized by $BrO^-$ to $NO_2^-$, and finally measured for $\delta^{15}N$ as $NO_2^-$ at 20 nmol using the denitrifier method. International $NH_4^+$ reference standards IAEA-N1, USGS25, and

USGS26 underwent the same preparation procedure as the soil KCl extracts and were used along with the $NO_3^-$ reference standards to correct for blanks and instrument drift. The precision of the $\delta^{15}N$-$NH_4^+$ analysis is ±0.5‰ (Yu and Elliott, 2018).

        $\delta^{15}N$ of NO collected in the TEA solution was measured following the method described in Yu and Elliott (2017). Briefly, the TEA collection samples were first neutralized with 12 N HCl to pH ~7, and then 10 to 20 nmol

of the collected product $NO_2^-$+$NO_3^-$ was converted to $N_2O$ using the denitrifier method. In light of the low $\delta^{15}N$ values of soil-emitted NO and the presence of $NO_2^-$ as the dominant collection product, a low $\delta^{15}N$ $NO_2^-$ isotopic standard (KNO2, RSIL20, USGS Reston; $\delta^{15}N$ = -79.6‰) was used together with the international $NO_3^-$ reference standards to calibrate the $\delta^{15}N$-NO analysis. Following the identical treatment principle, we prepared the isotopic standards in the same matrix (i.e., 20% TEA) as the collection samples and matched both the molar N amount and

injection volume (±5%) between the collection samples and the standards to minimize the blank interferences associated with the bacterial medium and the TEA solution. The precision and accuracy of the $\delta^{15}N$-NO analysis,



determined by repeated sampling of an analytical NO tank ($\delta^{15}$N-NO = -71.4‰) under diverse collection conditions, is ±1.1‰ (Yu and Elliott, 2017).

### 3 Results

Sixty-three NO collection samples were obtained from the incubation experiments. The NO collection efficiency calculated based on the measured $NO_2^-$+$NO_3^-$ concentration in the TEA solution and the theoretical concentration based on the measured net NO production rate (Yu and Elliott, 2017) was on average 99.1±3.7%. Out of the sixty-three collection samples, four samples had a NO collection efficiency lower than 95%. These samples were excluded from further data analysis and interpretation. The measured N concentrations, net NO production rates, and

isotope data from all the incubation experiments are available in Table S5 to Table S11 in the Supplement.

### 3.1 Anoxic incubation

During the anoxic incubation, soil $NO_3^-$ concentration decreased linearly from 49.3±0.1 µg N·g$^{-1}$ to 23.1±0.2 µg N·g$^{-1}$ (Fig. 2a), while $NO_2^-$ concentration increased linearly from 0.4±0.1 µg N·g$^{-1}$ to 6.9±0.1 µg N·g$^{-1}$ (Fig. 2b). The net NO production rate ($f_{NO-anoxic}$) increased progressively from the first sampling day (72±8 ng N·g$^{-1}$·h$^{-1}$) to

sampling day 5 and then stabilized at about 82 ng N·g$^{-1}$·h$^{-1}$ (Fig. 2c).

$\delta^{15}$N-$NO_3^-$ and $\delta^{15}$N-NO values increased from 4.7±0.3 to 38.7±1.5‰ and -44.7±0.3 to -22.8±2.2‰, respectively, over the anoxic incubation (Fig. 2d and 2f). The difference between $\delta^{15}$N-$NO_3^-$ and $\delta^{15}$N-NO values increased significantly from 49.4 to 59.5‰ toward the end of the incubation (Fig. 2d and 2f). Based on the closed-system Rayleigh model, the apparent N isotopic fractionation during $NO_3^-$ consumption was estimated to be

43.3±0.9‰ (Fig. S3 in the Supplement). $\delta^{15}$N-$NO_2^-$ was estimated for samples collected in the last three sampling days where $NO_2^-$ accounted for >15% of the $NO_3^-$+$NO_2^-$ pool. The estimated $\delta^{15}$N-$NO_2^-$ values were -6.9±3.7‰, -6.0±2.5‰, and -0.9±1.3‰, respectively (Fig. 2e). Although limited to the last three sampling days, $\delta^{15}$N-$NO_2^-$ was lower than $\delta^{15}$N-$NO_3^-$ by 33.6 to 37.9‰ (Fig. 2d and 2e), but was higher than the concurrently measured $\delta^{15}$N-NO values by a relatively constant offset of 21.5±0.7‰ (Fig. 2e and 2f). Surprisingly, both $\delta^{18}$O-$NO_3^-$ values (33.4±0.2

to 23.1±0.3‰) and $\Delta^{17}$O-$NO_3^-$ values (10.0±0.2 to 0.7±0.2‰) decreased progressively over the course of the anoxic incubation and were entirely decoupled from $\delta^{15}$N-$NO_3^-$ (Fig. 2g and 2h).

### 3.2 Oxic and hypoxic incubations

Over the oxic incubation, soil $NH_4^+$ concentration decreased linearly with increasing $NO_3^-$ concentration under all three $\delta^{15}$N-$NH_4^+$ treatments (Fig. 3a and 3b). In the hypoxic incubation, changes in $NH_4^+$ and $NO_3^-$ concentrations

were more limited, although the linear trends were still evident (Fig. 3a and 3b). Under both oxic and hypoxic conditions, the total concentration of soil $NH_4^+$ and $NO_3^-$ remained nearly constant over the entire incubations (i.e., variations < 4%), and soil $NO_2^-$ concentration was below the detection limit in both incubations. In the oxic incubation, $\delta^{15}$N-$NH_4^+$ values uniformly increased by 8.6 to 13.1‰ under all three $\delta^{15}$N-$NH_4^+$ treatments (Fig. 3e), while $\delta^{15}$N-$NO_3^-$ values varied distinctly, depending on the initial $\delta^{15}$N-$NH_4^+$ values (Fig. 3d). Specifically, $\delta^{15}$N-

$NO_3^-$ values increased by 7.8‰ and decreased by 10.9‰ under the high and low $\delta^{15}$N-$NH_4^+$ treatments, respectively,



and remained relatively constant under the intermediate $\delta^{15}N$-$NH_4^+$ treatment (Fig. 3d). Limited increases in $\delta^{15}N$-$NH_4^+$ values (<2‰) were observed under all three $\delta^{15}N$-$NH_4^+$ treatments in the hypoxic incubation (Fig. 3e). Correspondingly, variations in $\delta^{15}N$-$NO_3^-$ values were much smaller in the hypoxic incubation compared to those revealed in the oxic incubation (Fig. 3d). In both oxic and hypoxic incubations, $\delta^{18}O$-$NO_3^-$ (Fig. 3g) and $\Delta^{17}O$-$NO_3^-$

(Fig. 3h) values decreased progressively under all three $\delta^{15}N$-$NH_4^+$ treatments, although the rates of decrease were significantly higher in the oxic incubation (Fig. 3g and 3h).

The net NO production was significantly higher in the hypoxic incubation ($f_{NO\text{-}hypoxic}$; 9.0 to 10.4 ng N·$g^{-1}$·$h^{-1}$) than in the oxic incubation ($f_{NO\text{-}oxic}$; 7.1 to 8.5 ng N·$g^{-1}$·$h^{-1}$) (Fig. 3c). The measured $\delta^{15}N$-NO values ranged from -16.8±0.3 to -54.9±0.8‰ in the oxic incubation and from -21.3±0.0 to -51.4±0.4‰ in the hypoxic incubation (Fig.

3f). Pooling all the $\delta^{15}N$-NO measurements, we found that $\delta^{15}N$ values between $NH_4^+$ and NO differed from 58.9 to 70.7‰ across the three $\delta^{15}N$-$NH_4^+$ treatments in the oxic incubation and from 50.4 to 69.6‰ in the hypoxic incubation (Fig. 4). In both incubations, the largest difference was observed under the high $\delta^{15}N$-$NH_4^+$ treatment, while the smallest difference was observed under the low $\delta^{15}N$-$NH_4^+$ treatment. Under both oxic and hypoxic conditions, there was a significant linear relationship between the measured $\delta^{15}N$-NO and $\delta^{15}N$-$NH_4^+$ values from all

three $\delta^{15}N$-$NH_4^+$ treatments (Fig. 4). The slope of the linear relationship is 0.78±0.03 (± 1 SE) and 0.61±0.05 for the oxic and hypoxic incubations, respectively (Fig. 4).

### 3.3 Abiotic NO production

Addition of $NO_3^-$ or $NH_4^+$ to the sterilized soil did not result in detectable NO production under either oxic or anoxic condition. Immediate NO release was, however, triggered by $NO_2^-$ addition under anoxic conditions (Fig. 5a). The

abiotic NO production rate ($f_{NO\text{-}abiotic}$) reached a steady level of 83±5 ng N·$g^{-1}$·$h^{-1}$ several minutes after the $NO_2^-$ addition and then decreased exponentially to < 3 ng N·$g^{-1}$·$h^{-1}$ over the following 8 days (Fig. 5a). The natural logarithm of $f_{NO\text{-}abiotic}$ showed a linear relationship with time (Fig. 5b). The NO produced following the $NO_2^-$ addition had a $\delta^{15}N$ value of -17.8±0.4‰, giving rise to a $\delta^{15}N$ offset between $NO_2^-$ and NO of 19.2±0.5‰.

## 4 Discussion

Because interpretations of the results from the incubation experiments build upon each other, here we discuss the results from incubation of the sterilized soils (hereafter, abiotic incubation), anoxic incubation, and oxic/hypoxic incubations successively.

### 4.1 Reaction characteristics and N isotopic fractionation during abiotic NO production

The immediate release of NO upon the addition of $NO_2^-$ highlights the chemically unstable nature of $NO_2^-$ and the

critical role of chemical $NO_2^-$ reactions in driving soil NO emissions (Venterea et al., 2005; Lim et al., 2018). The strong linearity between $\ln(f_{NO\text{-}abiotic})$ and time (Fig. 5b) suggests apparent first-order kinetics for the abiotic NO production from $NO_2^-$ (Equations 2 and 3) (McKenney et al., 1990).

$$f_{NO\text{-}abiotic} = s_{abiotic} \times k_{abiotic} \times [NO_2^-]_t \qquad \text{Equation (2)}$$

$$[NO_2^-]_t = [NO_2^-]_0 e^{-k_{abiotic} \times t} \qquad \text{Equation (3)}$$





In Equations 2 and 3, $t$ is time; $k_{abiotic}$ is the pseudo-first order rate constant for $NO_2^-$ loss; $s_{abiotic}$ is the apparent stoichiometric coefficient for NO production from $NO_2^-$; and $[NO_2^-]_t$ and $[NO_2^-]_0$ are $NO_2^-$ concentration at time $t$ and $t=0$ in the sterilized soil, respectively. Combining Equations 2 and 3 and then log-transforming both sides yield:

$$\ln(f_{NO\text{-}abiotic}) = -k_{abiotic} \times t + \ln(s_{abiotic} \times k_{abiotic} \times [NO_2^-]_0)  \quad\quad \text{Equation (4)}$$

According to Equation 4, $k_{abiotic}$ and $s_{abiotic}$ are estimated using the slope and intercept of the linear regression of
$\ln(f_{NO\text{-}abiotic})$ versus time (Fig. 5b). Given $[NO_2^-]_0 = 8$ µg N·g⁻¹, $s_{abiotic}$ and $k_{abiotic}$ are estimated to be 0.52±0.05 (±SE) and 0.019±0.002 h⁻¹, respectively, suggesting that NO accounted for 52±5% of the reacted $NO_2^-$ during the abiotic incubation. The estimated $k_{abiotic}$ is within the range (i.e., 0.00055 to 0.73 h⁻¹) derived by a recent study based on soil samples spanning a wide range of pH values (3.4 to 7.2) (Lim et al., 2018). Based on the estimated $k_{abiotic}$, 97% of the added $NO_2^-$ was lost by the end of the abiotic incubation.

Several reaction pathways with distinct stoichiometry have been proposed for abiotic NO production from $NO_2^-$ in soils. Under acidic soil conditions, self-decomposition of $HNO_2$ produces NO and nitric acid ($HNO_3$) with a stoichiometric $HNO_2$-to-NO ratio ranging from 0.5 to 0.66 (i.e., 1 mole of $HNO_2$ produces 0.5 to 0.66 mole of NO) (Van Cleemput and Samater, 1995). Although at pH 5.7, $HNO_2$ constituted <1% of the $NO_2^-$+$HNO_2$ pool in this soil, $HNO_2$ decomposition can occur on acidic clay mineral surfaces, even though bulk soil pH is circumneutral
(Venterea et al., 2005). However, given the complete $NO_2^-$ consumption in the abiotic incubation, $HNO_2$ decomposition confined to acidic microsites could not account for all observed NO production. Under anoxic conditions, $NO_2^-$/$HNO_2$ can also be stoichiometrically reduced to NO by transition metals (e.g., Fe(II)) and diverse organic molecules (e.g., humic and fulvic acids, lignins, and phenols) in a process termed chemo-denitrification (Zhu-Baker et al., 2015). The produced NO from chemo-denitrification can undergo further reduction to form $N_2O$
and $N_2$ (Zhu-Baker et al., 2015). In addition, both $NO_2^-$ and NO in soil solution can be consumed as nitroso donors in abiotic nitrosation reactions, resulting in N incorporation into soil organic matter (Heil et al., 2016; Lim et al., 2018). Therefore, our observation that about half of the reacted $NO_2^-$ was recovered as NO may result from multiple competing $NO_2^-$ sinks, parallel NO-producing pathways, and possibly abiotic NO consumption in the sterilized soil. The other half of the reacted $NO_2^-$ that could not be accounted for by the measured NO was likely present in the
forms of $N_2O$, $N_2$, and/or nitrosated organic compounds in the soil.

        The observed $\delta^{15}N$ difference between $NO_2^-$ and NO (i.e., $^{15}\eta_{NO2/NO(abiotic)}$ = 19.2±0.5‰) likely reflects a combined N isotope effect for all of the competing NO production pathways during the abiotic incubation. While very little isotope data exist for abiotic $NO_2^-$ reactions in the literature, the measured $^{15}\eta_{NO2/NO(abiotic)}$ in this study is consistent with reported N isotope effects (i.e., 15 to 25‰) for abiotic $NO_2^-$ reduction by Fe(II) at similar $NO_2^-$
consumption rates as this study (0.02 to 0.05 h⁻¹) (Buchwald et al., 2016). On the other hand, the measured $^{15}\eta_{NO2/NO(abiotic)}$ is lower than the reported $\delta^{15}N$ offsets between $NO_2^-$ and $N_2O$ (i.e., $^{15}\eta_{NO2/N2O(abiotic)}$) for chemo-denitrification (24 to 29‰) (Jones et al., 2015; Wei et al., 2019). This seems to suggest that the observed abiotic NO production was mainly driven by chemo-denitrification and that accumulation of NO as an chemo-denitrification intermediate may explain why the observed $^{15}\eta_{NO2/N2O(abiotic)}$ was larger than the N isotope effect for Fe(II)-catalyzed
$NO_2^-$ reduction in previous batch experiments (Jones et al., 2015; Buchwald et al., 2016). Future studies adopting





simultaneous $\delta^{15}$N-NO and $\delta^{15}$N-N$_2$O measurements will be required to elucidate the role of NO as the N$_2$O precursor during chemo-denitrification.

It is important to note that the autoclaving is a harsh sterilization method and can substantially alter soil physical and chemical properties. For example, Buessecker et al. (2019) recently showed that autoclaved peat soil had 10-fold higher total fluorescence compared to non-sterilized controls, indicating dramatic increases in solubility and lability of organic molecules by autoclaving. Furthermore, autoclaving has also been shown to substantially increase abiotic N$_2$O production from NO$_2^-$-amended soils (Wei et al., 2019). Conversely, milder sterilization methods (e.g., gamma-irradiation) that presumably cause less alteration of soil properties may not completely inactivate biological NO production due to the high diversity of biological NO production pathways in soils (e.g., non-specific reactions catalyzed by extracellular enzymes) (Medinets et al., 2015). Further research is warranted to compare different sterilization methods for their effects on abiotic NO production and $^{15}\eta_{NO2/NO(abiotic)}$.

**4.2 Reaction reversibility between NO$_3^-$ and NO$_2^-$ and N isotope distribution between NO$_3^-$, NO$_2^-$, and NO during the anoxic incubation**

The measured $f_{NO\text{-anoxic}}$ (72 to 82 ng N·g$^{-1}$·h$^{-1}$) (Fig. 2c) is well within the range reported for anoxic soil incubations (e.g., 5 to 500 ng N·g$^{-1}$·h$^{-1}$) (Medinets et al., 2015), and is about 2/3 of the net consumption rate of NO$_3^-$+NO$_2^-$ during the anoxic incubation. That the majority of consumed NO$_3^-$+NO$_2^-$ was recovered as NO supports the emerging notion that NO can be the end product of denitrification once limitations on gas diffusion are lifted in soils (Russow et al., 2009; Loick et al., 2016). Applying the derived $k_{abiotic}$ and $s_{abiotic}$ in the abiotic incubation to the measured NO$_2^-$ concentrations under anoxic condition produced a range of $f_{NO\text{-abiotic}}$ from < 4 to 68 ng N·g$^{-1}$·h$^{-1}$ (Fig. S4 in the Supplement). While this modeled $f_{NO\text{-abiotic}}$ appears to contribute up to 80% of the measured $f_{NO\text{-anoxic}}$ (Fig. S4 in the Supplement), $f_{NO\text{-anoxic}}$ was high and remained stable even without any significant accumulation of NO$_2^-$ in the soil (Fig. 2b and 2c), suggesting that $k_{abiotic}$ was likely overestimated in the abiotic incubation (see above). Assuming that net biological NO production was maintained at the level of $f_{NO\text{-anoxic}}$ measured during the first sampling event and that $s_{abiotic}$ was constant and equal to 0.52, a back-of-the-envelope calculation based on the difference in $f_{NO\text{-anoxic}}$ between the first and last sampling events and the NO$_2^-$ concentration measured at the end of the anoxic incubation indicates that $k_{abiotic}$ was likely on the order of 0.0027 h$^{-1}$, or about 7 times lower than the $k_{abiotic}$ derived in the abiotic incubation. Although qualitative, this calculation suggests a minor contribution of abiotic NO production to the measured $f_{NO\text{-anoxic}}$ (<12%; Fig. S4 in the Supplement).

The large increases in $\delta^{15}$N-NO$_3^-$ and $\delta^{15}$N-NO values over the anoxic incubation (Fig. 2d and 2f) are congruent with strong N isotopic fractionations during microbial denitrification (Mariotti et al., 1981; Granger et al., 2008). However, the observed net isotope effect for NO production from NO$_3^-$ (i.e., $^{15}\eta_{NO3/NO}$; 49.4 to 59.5‰) is larger than the apparent N isotope effect for NO$_3^-$ consumption (43.3±0.9‰) (Fig. S3 in the Supplement). The large magnitude and increasing pattern of $^{15}\eta_{NO3/NO}$, together with the accumulation of NO$_2^-$ in the soil, point to complexity beyond single-step isotopic fractionations and highlight the need to carefully examine fractionation mechanisms for all intermediate steps leading to net NO production (i.e., NO$_3^-$ to NO$_2^-$, NO$_2^-$ to NO, and NO to N$_2$O). Moreover, it is surprising that both $\delta^{18}$O-NO$_3^-$ and $\Delta^{17}$O-NO$_3^-$ values decreased over the anoxic incubation (Fig. 2g and 2h). Interestingly, similar decreasing trends in $\delta^{18}$O-NO$_3^-$ values (e.g., up to 4‰ over 25 h) have been



reported by Lewicka-Szczebak et al. (2014) for two anoxically incubated agricultural soils amended with a high-$\delta^{18}$O Chilean $NO_3^-$ fertilizer similar to ours (i.e., $\delta^{18}$O-$NO_3^-$ = 56‰), although $\Delta^{17}$O-$NO_3^-$ was not reported in this

previous study. The decreasing $\delta^{18}$O-$NO_3^-$ values, observed here and by Lewicka-Szczebak et al. (2014), appear to contradict the well-established paradigm that variations in $\delta^{15}$N-$NO_3^-$ and $\delta^{18}$O-$NO_3^-$ values follow a linear trajectory with a slope of 0.5 to 1 during dissimilatory $NO_3^-$ reduction (Granger et al., 2008). Furthermore, as $\Delta^{17}$O-$NO_3^-$ is in theory not altered by microbial denitrification – a mass-dependent fractionation process (Michalski et al., 2004; Yu and Elliott, 2018), the decreasing $\Delta^{17}$O-$NO_3^-$ values observed in this study indicate that processes capable of diluting

or erasing the $\Delta^{17}$O signal may occur concurrently with denitrification during the anoxic incubation. Importantly, if this dilution or removal of the $\Delta^{17}$O signal was accompanied by N isotopic fractionations, there may be cascading effects on the distribution of N isotopes between $NO_3^-$, $NO_2^-$, and $NO$.

       The decreasing $\delta^{18}$O-$NO_3^-$ and $\Delta^{17}$O-$NO_3^-$ values could be potentially explained by an O isotope equilibration between $NO_3^-$ and soil $H_2O$, catalyzed either chemically or biologically via a reversible reaction

between $NO_3^-$ and $NO_2^-$ (Granger and Wankel, 2016). However, it has been shown in controlled laboratory experiments that dissimilatory $NO_3^-$ reduction catalyzed by bacterial nitrate reductase (NAR) is irreversible at the enzyme level (Treibergs and Granger, 2017) and that abiotic O isotope exchange between $NO_3^-$ and $H_2O$ is extremely slow (half-life >$10^9$ y at 25°C and pH 7) and therefore irrelevant under natural soil conditions (Kaneko and Poulson, 2013). Although fungi use a distinct enzyme system for denitrification (Shoun et al., 2012), there is no

evidence for enzymatic reversibility of fungal NAR in the literature. Furthermore, by converting $NH_4^+$ and $NO_2^-$ simultaneously to $N_2$ and $NO_3^-$, anaerobic $NH_4^+$ oxidation (anammox) could dilute the $\Delta^{17}$O signal by producing $NO_3^-$ with $\Delta^{17}$O=0 (Brunner et al., 2013). However, due to the low indigenous $NH_4^+$ concentration, anammox is considered not pertinent during the anoxic incubation. Given the complete recovery of $NO_3^-$ concentrations and isotopes in the control experiments (Table S1 and Table S2 in the Supplement), as well as the significantly increased

$\delta^{15}$N-$NO_3^-$ values during the anoxic incubation, we excluded $NO_3^-$ production from aerobic $NH_4^+$ oxidation as a possible explanation for the observed declines in $\delta^{18}$O-$NO_3^-$ and $\Delta^{17}$O-$NO_3^-$ values.

       Therefore, having ruled out the above possibilities led us to postulate that the decreasing $\delta^{18}$O-$NO_3^-$ and $\Delta^{17}$O-$NO_3^-$ values may result from anaerobic $NO_2^-$ oxidation mediated by NOB in the soil. The enzyme catalyzing $NO_2^-$ oxidation to $NO_3^-$ in NOB – $NO_2^-$ oxidoreductase (NXR) – is metabolically versatile and has been shown to

catalyze $NO_3^-$ reduction under anoxic conditions by operating in reverse (Friedman et al., 1986; Freitag et al., 1987; Bock et al., 1988; Koch et al., 2015). Moreover, during NXR-catalyzed $NO_2^-$ oxidation, the required O atom originates from $H_2O$ molecules (Reaction 1), so that $NO_2^-$ can in theory be oxidized to $NO_3^-$ without the presence of $O_2$ by donating electrons to redox-active intracellular components (Wunderlich et al., 2013) or alternative electron acceptors in niche environments (Babbin et al., 2017).

$$NO_3^- + 2H^+ + 2e^- \Leftrightarrow H_2O + NO_2^-$$        Reaction 1

In a denitrifying environment, anaerobic oxidation of denitrification-produced $NO_2^-$ back to $NO_3^-$ (i.e., $NO_2^-$ re-oxidation) can dilute $\delta^{18}$O-$NO_3^-$ and $\Delta^{17}$O-$NO_3^-$ values by incorporating a 'new' O atom from $H_2O$ into the reacting $NO_3^-$ pool (Reaction 1) (Granger and Wankel, 2016). Under acidic and circumneutral pH conditions, this dilution effect can be further enhanced by chemically- and perhaps biologically-catalyzed O isotope equilibration between



$NO_2^-$ and $H_2O$ (Casciotti et al., 2007; Buchwald and Casciotti, 2010), which effectively erase the isotopic imprints of denitrification on $NO_2^-$ prior to its re-oxidation. The reversibility of NXR and its direct control on O isotopes in $NO_3^-$ have been convincingly demonstrated by Wunderlich et al. (2013) using a pure culture of *Nitrobacter vulgaris*. By incubating *N. vulgaris* in a $NO_3^-$ solution under anoxic conditions, Wunderlich et al. (2013) showed that $NO_2^-$ was produced in the solution by *N. vulgaris* and that *N. vulgaris* promoted incorporation of amended $^{18}O$-$H_2O$ labels into

$NO_3^-$ through a re-oxidation of the accumulated $NO_2^-$ (Wunderlich et al., 2013).

Importantly, there is mounting evidence from the marine N cycle community that $NO_2^-$ re-oxidation plays a critical role in the N isotope partitioning between $NO_3^-$ and $NO_2^-$. At the process scale, $NO_2^-$ re-oxidation co-occurring with dissimilatory $NO_3^-$ reduction can lead to a large $\delta^{15}N$ difference between $NO_3^-$ and $NO_2^-$ beyond what would be expected to result from $NO_3^-$ reduction alone (Gaye et al., 2013; Dale et al., 2014; Dähnke and Thamdrup,

2015; Peters et al., 2016; Martin and Casciotti, 2017; Buchwald et al., 2018). This large $\delta^{15}N$ difference is thought to arise from a rare, but intrinsic, inverse kinetic isotope effect associated with $NO_2^-$ re-oxidation (e.g., -13‰) (Casciotti et al., 2009). As such, in a net denitrifying environment, $NO_2^-$ re-oxidation functions as an apparent branching pathway along the sequential reduction of $NO_3^-$, preferentially re-oxidizing $^{15}NO_2^-$ back to $NO_3^-$. At the enzyme scale, the bidirectional NXR enzyme has been proposed to catalyze intracellular coupled $NO_3^-$ reduction and

$NO_2^-$ oxidation (i.e., bidirectional interconversion of $NO_3^-$ and $NO_2^-$), facilitating expression of an equilibrium N isotope effect between $NO_3^-$ and $NO_2^-$ (Reaction 2) (Wunderlich et al., 2013; Kemeny et al., 2016).

$$^{14}NO_2^- + {}^{15}NO_3^- \Leftrightarrow {}^{15}NO_2^- + {}^{14}NO_3^-$$    Reaction 2

Evidence from pure culture studies of anammox bacteria carrying the NXR enzyme (Brunner et al., 2013) and theoretical quantum calculations (Casciotti, 2009) suggests that this N isotope equilibration favors partitioning of

$^{14}N$ into $NO_2^-$ with an equilibrium isotope effect ranging from -50 to -60‰ (negative sign is used to denote that this N isotope equilibration partitions $^{14}N$ to the left side of Reaction 2). This NXR-catalyzed $NO_3^-$/$NO_2^-$ interconversion was invoked to explain the extremely low $\delta^{15}N$-$NO_2^-$ values relative to $\delta^{15}N$-$NO_3^-$ (up to 90‰) in the surface Antarctic ocean, where aerobic $NO_2^-$ oxidation is inhibited by low nutrient availability (Kemeny et al., 2016). Hypothetically, if expressed at either the process or the enzyme level, the N isotope effect for $NO_2^-$ re-oxidation

could propagate into denitrification-produced NO, giving rise to an increased $\delta^{15}N$ difference between $NO_3^-$ and NO ($^{15}\eta_{NO3/NO}$).

To test whether $NO_2^-$ re-oxidation can explain the observed declines in $\delta^{18}O$-$NO_3^-$ and $\Delta^{17}O$-$NO_3^-$ values and $\delta^{15}N$ distribution between $NO_3^-$, $NO_2^-$, and NO, we modified an isotopologue-specific (i.e., $^{14}N$, $^{15}N$, $^{16}O$, $^{17}O$, and $^{18}O$) numerical model previously described by Yu and Elliott (2018) to simulate co-occurring denitrification and

$NO_2^-$ re-oxidation in two steps. Without a clear identification of the alternative electron acceptors that coupled with anaerobic $NO_2^-$ oxidation in the studied soil, we followed the reaction scheme proposed by Wunderlich et al. (2013) and Kemeny et al. (2016) (Reaction 1) to parameterize the NXR-catalyzed $NO_2^-$ re-oxidation as the backward reaction of a dynamic equilibrium between $NO_3^-$ and $NO_2^-$ (Fig. 6) – that is, the NXR-catalyzed $NO_2^-$ re-oxidation (backward reaction) is balanced by an NXR-catalyzed $NO_3^-$ reduction (forward reaction), leading to no net $NO_2^-$

oxidation or $NO_3^-$ reduction in the soil. Importantly, this representation is consistent with the observation that both $NO_3^-$ consumption and $NO_2^-$ accumulation followed a pseudo-zero order kinetics over the anoxic incubation (Fig. 2a





and 2b), which implies no net contribution from the $NO_3^-$/$NO_2^-$ interconversion. Given previous findings that the NXR-catalyzed O exchange between $NO_3^-$ and $NO_2^-$ depends on $NO_2^-$ availability (Wunderlich et al., 2013), the backward $NO_2^-$ re-oxidation was assumed to be first order (with respect to $NO_2^-$), defined by a first order rate constant, $k_{NXR(b)}$. With respect to the O isotope equilibration between $H_2O$ and the reacting $NO_2^-$ pool, we considered two extreme case scenarios: (1) no exchange and (2) complete exchange. In the "no exchange" scenario, the imprints of denitrification on $\delta^{18}O$-$NO_2^-$ and $\Delta^{17}O$-$NO_2^-$ values are preserved, such that only one $H_2O$-derived O atom is incorporated into $NO_3^-$ with each $NO_2^-$ molecule being re-oxidized (Reaction 1). In the "complete exchange" scenario, $\delta^{18}O$ and $\Delta^{17}O$ values of $NO_2^-$ always reflect those of soil $H_2O$ ($\delta^{18}O$-$H_2O\approx$-10‰, $\Delta^{17}O$-$H_2O$=0‰) (Fig. 6), and therefore all three O atoms in $NO_3^-$ produced from $NO_2^-$ re-oxidation originate from $H_2O$. Furthermore, we considered both abiotic NO production and denitrification as the source of NO during the anoxic incubation (Fig. 6). To account for the potential overestimation in $k_{abiotic}$ (see above), we used a reduced $k_{abiotic}$ (0.0027 h$^{-1}$) to model abiotic NO production from $NO_2^-$, while $s_{abiotic}$ and $^{15}\eta_{NO2/NO(abiotic)}$ were fixed at 0.52 and 19.2‰, respectively. With respect to $\delta^{15}N$ of denitrification-produced NO, we assumed that NIR-catalyzed $NO_2^-$ reduction to NO and NOR-catalyzed NO reduction to $N_2O$ were each associated with a kinetic N isotope effect ($^{15}\eta_{NIR}$ and $^{15}\eta_{NOR}$). The closed-system Rayleigh equation was then used to simulate the coupled NO production and reduction in denitrification at each model time interval (Lewicka-Szczebak et al. 2014). Detailed model derivation and formulation are provided in the Supplement (Text S3.1).

With this model of co-occurring denitrification and $NO_2^-$ re-oxidation, we first solved for the rates of denitrifier-catalyzed $NO_3^-$ ($R_{NAR}$), $NO_2^-$ ($R_{NIR}$), and NO ($R_{NOR}$) reductions and $k_{NXR(b)}$ (4 unknowns) using the measured $NO_3^-$ and $NO_2^-$ concentrations, $f_{NO\text{-}anoxic}$, and $\Delta^{17}O$-$NO_3^-$ values (4 measured variables). This first modeling step was robustly constrained by the measured $\Delta^{17}O$-$NO_3^-$, which essentially functions as a $^{15}NO_3^-$ tracer (Yu and Elliott, 2018) and is therefore particularly sensitive to $NO_2^-$ re-oxidation. In the second modeling step, the measured $\delta^{15}N$-$NO_3^-$, $\delta^{15}N$-$NO_2^-$, and $\delta^{15}N$-NO values (3 measured variables) were used to optimize the kinetic N isotope effects for NAR-catalyzed $NO_3^-$ reduction ($^{15}\eta_{NAR}$), $^{15}\eta_{NIR}$, $^{15}\eta_{NOR}$, and the equilibrium N isotope effect for NXR-catalyzed $NO_3^-$/$NO_2^-$ interconversion ($^{15}\eta_{NXR(eq)}$) (Reaction 2; Fig. 6) (4 unknowns). This modeling system is under-determined (number of measured variables < number of unknowns) and thus cannot be solved uniquely. Thus, instead of definitively solving for the four unknown isotope effects, we explored their best combination to fit the measured $\delta^{15}N$ values of $NO_3^-$, $NO_2^-$, and NO. Specifically, to reduce the number of unknowns for model optimization, $^{15}\eta_{NAR}$ and $^{15}\eta_{NXR(eq)}$ were treated as known values, and $^{15}\eta_{NIR}$ and $^{15}\eta_{NOR}$ were solved by mapping through the entire space of $^{15}\eta_{NAR}$ and $^{15}\eta_{NXR(eq)}$ (at a resolution of 1‰), defined by their respective widest range of possible values. We used a range of 5 to 55‰ for $^{15}\eta_{NAR}$, consistent with a recent compilation based on soil incubations and denitrifier pure cultures (Denk et al., 2017). Given the existing observational and theoretical constraints (Casciotti, 2009; Brunner et al., 2013), a range of -60 to 0‰ was assigned to $^{15}\eta_{NXR(eq)}$, which is equivalent to the argument that the impact of $NO_3^-$/$NO_2^-$ interconversion on the N isotope distribution between $NO_3^-$ and $NO_2^-$ can vary from null to a strong partitioning of $^{14}N$ to $NO_2^-$. We further defined the lower 2.5th percentile of the error-weighted residual sum of squares (RSS) between simulated and measured $\delta^{15}N$ values of $NO_3^-$, $NO_2^-$, and





NO as the threshold for selection of the best-fit models. Detailed information regarding model optimization can be found in the Supplement (Text S3.2).

Results from the first modeling step are summarized in Table 1 and the best-fit models were plotted in Fig. 2 to compare with the measured data. Because the NXR-catalyzed $NO_3^-/NO_2^-$ interconversion was assumed to result in no change in $NO_3^-$ and $NO_2^-$ concentrations, $R_{NAR}$ (0.158 µg N·g$^{-1}$·h$^{-1}$), $R_{NIR}$ (0.112 µg N·g$^{-1}$·h$^{-1}$), and $R_{NOR}$ (0.039 µg N·g$^{-1}$·h$^{-1}$) can be well-described by zero-order kinetics and are not sensitive to model scenarios for O exchange between $NO_2^-$ and $H_2O$ (Table 1). Moreover, the observed $NO_2^-$ accumulation and $f_{NO\text{-anoxic}}$ dynamics can be well-

reproduced using the modeled denitrification rates and the downward adjustment of $k_{abiotic}$ (Fig. 2b and 2c). $k_{NXR(b)}$ was estimated to be 0.64 h$^{-1}$ and 0.25 h$^{-1}$ under the "no exchange" and "complete exchange" scenarios, respectively (Table 1). Under both scenarios, the simulated $\Delta^{17}O\text{-}NO_3^-$ values exhibit a characteristic decreasing trend and are in excellent agreement with measured $\Delta^{17}O\text{-}NO_3^-$ values (Fig. 2h). The larger $k_{NXR(b)}$ under the "no exchange" scenario is expected and can be explained by the faster back reaction (i.e., $NO_2^-$ re-oxidation) required to reproduce the

observed dilution of $\Delta^{17}O\text{-}NO_3^-$, because only one "new" O atom is incorporated into $NO_3^-$ with each $NO_2^-$ molecule being re-oxidized. Although the measured $\delta^{18}O\text{-}NO_3^-$ values did not provide quantitative constraints for the model optimization, the isotopologue-specific model with the optimized denitrification rates and $k_{NXR(b)}$ was run forward to test whether the decreasing $\delta^{18}O\text{-}NO_3^-$ values can also be possibly explained by co-occurring denitrification and $NO_2^-$ re-oxidation (details are provided in Text S4 in the Supplement). The results showed that $NO_3^-$ reduction

(acting to increase $\delta^{18}O\text{-}NO_3^-$ values) and $NO_2^-$ re-oxidation (acting to decrease $\delta^{18}O\text{-}NO_3^-$ values) have counteracting effects on the forward-modeled $\delta^{18}O\text{-}NO_3^-$ (Fig. S2 in the Supplement) and that the decreasing trend in $\delta^{18}O\text{-}NO_3^-$ values can be well-reproduced under both "no exchange" and "complete exchange" scenarios with a reasonable assumption on the net O isotope effects for denitrification and $NO_2^-$ re-oxidation (Fig. S2; see Text S4 in the Supplement) (Granger and Wankel, 2016). Therefore, although $k_{NXR(b)}$ cannot be definitively quantified in this

study due to the unknown degree of O exchange between $NO_2^-$ and $H_2O$, these simulation results provide confidence in our hypothesis that the observed decreases in $\delta^{18}O\text{-}NO_3^-$ and $\Delta^{17}O\text{-}NO_3^-$ values were driven by the reversible action of the NXR enzyme. It is important to note that the estimated $k_{NXR(b)}$ is fairly large even under the "complete exchange" scenario. Based on the $NO_2^-$ concentration measured at the end of the anoxic incubation (6.9 µg N·g$^{-1}$), a $k_{NXR(b)}$ of 0.25 h$^{-1}$ would require a $NO_2^-$ re-oxidation rate (1.7 µg N·g$^{-1}$·h$^{-1}$) that is one order of magnitude higher

than the estimated $R_{NAR}$ and $R_{NIR}$. However, the inferred maximum $NO_2^-$ re-oxidation rate under either model scenario (1.7 to 4.4 µg N·g$^{-1}$·h$^{-1}$) is still within the reported range for aerobic $NO_2^-$ oxidation in agricultural soils (e.g., up to 6-7 µg N·g$^{-1}$·h$^{-1}$) (Taylor et al., 2019), indicative of high NOB activity even under anoxic conditions (Koch et al., 2015).

      Based on the modeled denitrification rates and $k_{NXR(b)}$, the best-fit $^{15}\eta_{NXR(b)}$ was confined to a narrow range

from -40 to -35‰ (Fig. 7a and 7b) and was not sensitive to model scenarios for O equilibration between $NO_2^-$ and $H_2O$ (Fig. 8b). While the best-fit $^{15}\eta_{NAR}$ and $^{15}\eta_{NXR(b)}$ were positively correlated, especially under the "complete exchange" scenario (Fig. 7a and 7b), the best-fit $^{15}\eta_{NAR}$ spanned a wide range (5 to 45‰) and was significantly lower under the "no exchange" scenario (RSS-weighted mean: 19‰) relative to the "complete exchange" scenario (RSS-weighted mean: 30‰) (Fig. 8a). On the other hand, the best-fit $^{15}\eta_{NIR}$ (15 to 22‰) and $^{15}\eta_{NOR}$ (-8 to 2‰) did



not vary substantially and were similar between the two model scenarios (Fig. 7c to 7d; Fig. 8c and 8d). Under both model scenarios, the measured $\delta^{15}N$-$NO_3^-$, $\delta^{15}N$-$NO_2^-$, and $\delta^{15}N$-$NO$ values can be well-simulated using the RSS-weighted mean $^{15}\eta$ values from the best-fit models (Fig. 2d to 2f). Specifically, the modeled difference between $\delta^{15}N$-$NO_3^-$ and $\delta^{15}N$-$NO_2^-$ values increased from about 29‰ at the beginning of the incubation to about 38‰ at the end of the incubation (Fig. 2d and 2e), whereas a constant $\delta^{15}N$ offset of about 20‰ was revealed between the

modeled $\delta^{15}N$-$NO_2^-$ and $\delta^{15}N$-$NO$ values (Fig. 2e and 2f). Therefore, the modeled $^{15}\eta$ values and $\delta^{15}N$-$NO_2^-$ dynamics reveal important new information for understanding the increasing $^{15}\eta_{NO3/NO}$ over the anoxic incubation. During the early phase of the incubation, the N isotope partitioning between $NO_3^-$, $NO_2^-$, and $NO$ was mainly controlled by denitrification and its associated isotope effects (i.e., $^{15}\eta_{NAR}$, $^{15}\eta_{NIR}$, and $^{15}\eta_{NOR}$). With the increasing accumulation of $NO_2^-$ in the soil, the dominant control on the $\delta^{15}N$ distribution shifted to the N isotope exchange

between $NO_3^-$ and $NO_2^-$, so that the difference between the $\delta^{15}N$-$NO_3^-$ and $\delta^{15}N$-$NO_2^-$ values was primarily determined by $^{15}\eta_{NXR(eq)}$ (-40 to -35‰). The revealed positive correlation between the best-fit $^{15}\eta_{NAR}$ and $^{15}\eta_{NXR(b)}$ (Fig. 7a and 7b) and the significantly lower $^{15}\eta_{NAR}$ under the "no exchange" scenario (Fig. 8a) essentially reflect a trade-off between $^{15}\eta_{NAR}$ and $^{15}\eta_{NXR(b)}$ in controlling the $\delta^{15}N$ difference between $NO_3^-$ and $NO_2^-$ – that is, when the interconversion between $NO_3^-$ and $NO_2^-$ is fast and the magnitude of $^{15}\eta_{NXR(eq)}$ is large (i.e., very negative), only a

small $^{15}\eta_{NAR}$ is required to sustain the large $\delta^{15}N$ difference between $NO_3^-$ and $NO_2^-$ over the course of the anoxic incubation.

The estimated $^{15}\eta_{NXR(eq)}$ from the best-fit models is higher (i.e., closer to zero) than those derived from theoretical calculations and pure culture studies (-50 to -60‰) (Casciotti, 2009; Brunner et al., 2013). Given the heterogeneous distribution of substrates in soils, the lower absolute magnitude of the best-fit $^{15}\eta_{NXR(eq)}$ may be due to

the partial rate limitation by transport of $NO_2^-$/$NO_3^-$ to the active site of NXR. As such, the best-fit $^{15}\eta_{NXR(eq)}$ should provide a conservative estimate of the intrinsic equilibrium isotope effect. Thus, the results from the anoxic incubation underscore the important, yet previously unrecognized, role of the reversible $NO_3^-$/$NO_2^-$ interconversion in controlling the $\delta^{15}N$ dynamics of soil $NO_3^-$ and its denitrification products. Substantial re-oxidation of $NO_2^-$ under anoxic conditions seems paradoxical, but is underpinned by the increasingly recognized high degree of metabolic

versatility of NOB, including simultaneous oxidation of an organic substrate and $NO_2^-$, as well as parallel use of $NO_3^-$ and $O_2$ as electron acceptors (Koch et al., 2015). In the absence of $O_2$, few electron acceptors exist at common environmental pH that have a higher redox potential than the $NO_3^-$/$NO_2^-$ pair (Wunderlich et al., 2013; Babbin et al., 2017). It is therefore likely that NOB would gain energy by performing the intracellular coupled oxidation of $NO_2^-$ and reduction of $NO_3^-$ to survive periods of $O_2$ deprivation. Although anaerobic $NO_2^-$ oxidation until now has been

conclusively shown only in anoxic ocean water columns (Sun et al., 2017; Babbin et al., 2017) and aquatic sediments (Wunderlich et al., 2013), soils host a huge diversity of coexisting NOB (Le Roux et al., 2016) and the physiological flexibility of NOB beyond aerobic $NO_2^-$ oxidation may contribute to the unexpected higher abundances and activities of NOB relative to AOB and AOA in agricultural soils (Høberg et al., 1996; Ke et al., 2013). Using the modified isotopologue-specific model, we demonstrate the possibility that large $^{15}\eta_{NAR}$ can be an

artifact of an isotopic equilibrium between $NO_3^-$ and $NO_2^-$, occurring in connection with the bifunctional NXR enzyme. Therefore, effective expressions of $^{15}\eta_{NXR(eq)}$ in concurrence with $^{15}\eta_{NAR}$ may explain why $^{15}\eta_{NAR}$ estimated





by some anoxic soil incubations (e.g., 25 to 65‰) are far larger than those reported by studies of denitrifying and $NO_3^-$-reducing bacterial cultures (e.g., 5 to 30‰) (Denk et al., 2017) and why the slope of $\delta^{18}O$-$NO_3^-$ versus $\delta^{15}N$-$NO_3^-$ values during denitrification in many field studies was not constant and rarely close to unity as observed in

pure denitrifying cultures (Granger and Wankely, 2016). Indeed, evidence for a reversible enzymatic pathway linking $NO_3^-$ and $NO_2^-$ under anoxic conditions has already been documented in previous soil studies (e.g., Kool et al., 2011; Lewicka-Szcebak et al., 2014), implying its wide occurrence in soils. More studies using soils from a broad range of environments are needed to pinpoint the exact mechanisms by which $NO_2^-$ can be anaerobically oxidized in soils. To that end, $\Delta^{17}O$-$NO_3^-$ can be used as a powerful benchmark for disentangling co-occurring $NO_3^-$

reduction and $NO_2^-$ re-oxidation.

The best-fit $^{15}\eta_{NIR}$ (15 to 22‰) falls within the range derived in anoxic soil incubations (11 to 33‰) (Mariotti et al., 1982) and is consistent with results based on denitrifying bacteria carrying copper-containing NIR (22‰) (Martin and Casciotti, 2016). Under both model scenarios, the best-fit $^{15}\eta_{NOR}$ (-8 to 2‰) is relatively small and more normal than the bulk N isotope effect for NO reduction to $N_2O$ catalyzed by purified fungal NOR

(P450nor) (-14‰) (Yang et al., 2014). During P450nor-catalyzed NO reduction, two NO molecules are sequentially bonded to the Fe active site of P450nor and the observed inverse isotope effect was proposed to arise from a reversible bonding of the first NO molecule (Yang et al., 2014). To date, the N isotope effect for NO reduction catalyzed by bacterial NORs has not yet been quantified. Unlike P450nor, which contains only a single heme Fe at the active site, the active site of bacterial NORs has two Fe atoms (i.e., binuclear center). Therefore, three classes of

mechanisms have been proposed for the two-electron reduction of NO by bacterial NORs, including sequential bonding of two NO molecules to either Fe catalytic center and simultaneous bonding of two NO molecules to both Fe centers (Kuypers et al., 2018; Lehnert et al., 2018). Although the precise catalytic mechanism remains uncertain, site-specific measurements of N isotopes in $N_2O$ (i.e., $N_2O$ isotopomers) produced from denitrifying bacteria indicate similar magnitude for isotopic fractionations during the reduction of two NO molecules, in support of the

simultaneous binding theory (Sutka et al., 2006; Yamazaki et al., 2014). Thus, if the bulk N isotope effect for bacterial NO reduction is higher than that for fungal NO reduction, the best-fit $^{15}\eta_{NOR}$ may reflect a mixed contribution of bacteria and fungi to NO consumption during the anoxic incubation. Alternatively, the model-inferred $^{15}\eta_{NOR}$ might reflect a balance between enzymatic and diffusion isotope effects, as has been previously demonstrated for $N_2O$ reduction in soil denitrification (Lewicka-Szczebak et al., 2014). Because diffusion would be

expected to have a small and normal kinetic isotope effect, if $NO_2^-$ reduction was limited by NO diffusion out of soil denitrifying sites, the estimated $^{15}\eta_{NOR}$ would be shifted toward the isotope effect for NO diffusion. Diffusion might be particularly important in this study due to the flow-through condition during the anoxic incubation and the low solubility of NO, both of which favor gas diffusion while preventing re-entry of escaped NO to denitrifying cells. Thus, the small $^{15}\eta_{NOR}$ inferred from the best-fit models is likely a combination of diverse NO reduction pathways in

this agricultural soil, as well as limited expression of enzymatic isotope effects imposed by NO diffusion. Regardless, the empirical finding of this study suggests that due to the small $^{15}\eta_{NOR}$, the bulk $\delta^{15}N$ values of denitrification-produced $N_2O$ should not be significantly altered by accumulation and diffusion of NO during denitrification.





### 4.3 NO source contribution and N isotope effects for NO production from $NH_4^+$ oxidation under oxic and hypoxic conditions

The coupled decrease in $NH_4^+$ concentrations and increase in $NO_3^-$ concentrations (Fig. 3a and 3b) indicate active nitrification in both oxic and hypoxic incubations. Moreover, the two oxidation steps of nitrification were tightly coupled, resulting in no accumulation of $NO_2^-$ in the soil. Because $NO_3^-$ produced from nitrification has a zero $\Delta^{17}O$ value, the active nitrification was also reflected in the progressive dilution of $\Delta^{17}O$-$NO_3^-$ under both oxic and hypoxic conditions (Yu and Elliott, 2018). Based on the measured concentrations and isotopic composition of $NH_4^+$ and $NO_3^-$, the isotopologue-specific model previously developed by Yu and Elliott (2018) was used to estimate the rates and net N isotope effects of net mineralization ($R_{OrgN/NH4}$ and $^{15}\eta_{OrgN/NH4}$), gross $NH_4^+$ oxidation to $NO_3^-$ ($R_{NH4/NO3}$ and $^{15}\eta_{NH4/NO3}$), and gross $NO_3^-$ consumption ($R_{NO3comp}$ and $^{15}\eta_{NO3comp}$) during the oxic and hypoxic incubations. As have been discussed above, this numerical model relies on the conservative nature of $\Delta^{17}O$-$NO_3^-$ and its powerful applications in tracing co-occurring nitrification and $NO_3^-$ consumption (consisting of $NO_3^-$ immobilization and denitrification in this case) (Yu and Elliott, 2018). Detailed model derivation, formulation, and optimization have been documented in Yu and Elliott (2018) and are also briefly summarized in Text S5 in the Supplement. The modeling results based on the low $\delta^{15}N$-$NH_4^+$ treatment in the oxic incubation were reported by Yu and Elliott (2018). Here, we used data from all three $\delta^{15}N$-$NH_4^+$ treatments to more robustly constrain the N transformation rates and net N isotope effects for each incubation experiment (i.e., oxic and hypoxic).

The modeling results were summarized in Table 2. Excellent agreement was obtained between the observed and simulated concentrations and isotopic composition of $NH_4^+$ and $NO_3^-$ for both oxic and hypoxic incubations (Fig. 3). $R_{NH4/NO3}$ can be well described by zero order kinetics and was estimated to be 0.46 µg N·$g^{-1}$·$h^{-1}$ and 0.11 µg N·$g^{-1}$·$h^{-1}$ for the oxic and hypoxic incubations, respectively (Table 2). The lower $R_{NH4/NO3}$ in the hypoxic incubation indicates that nitrification was limited by low $O_2$ availability. Under both oxic and hypoxic conditions, oxidation of $NH_4^+$ to $NO_3^-$ was associated with a large $^{15}\eta_{NH4/NO3}$ (23 to 28‰; Table 2), consistent with the N isotope effects for $NH_3$ oxidation in pure cultures of AOB and AOA (e.g., 13 to 41‰) (Mariotti et al., 1981; Casciotti et al., 2003; Santoro et al., 2011). On the other hand, the estimated $R_{OrgN/NH4}$ and $R_{NO3comp}$ were low and not significantly different between the two incubation experiments (Table 2). Nevertheless, while $R_{NO3comp}$ was only 16% of $R_{NH4/NO3}$ in the oxic incubation, $R_{NO3comp}$ accounted for a much larger fraction (63%) of $R_{NH4/NO3}$ in the hypoxic incubation, mainly due to the reduced $R_{NH4/NO3}$ under the low $O_2$ condition. Due to the low magnitude of $R_{OrgN/NH4}$ and $R_{NO3comp}$, the estimated $^{15}\eta_{OrgN/NH4}$ and $^{15}\eta_{NO3comp}$ are associated with large errors and not significantly different from zero (Table 2).

By using three isotopically different $NH_4^+$ fertilizers in parallel treatments, we are able to quantify the fractional contribution of $NH_4^+$ oxidation to the measured net NO production ($f_{NH4}$). Specifically, if NO was exclusively produced from soil $NH_4^+$, we would expect to see a constant $\delta^{15}N$ difference between $NH_4^+$ and NO across the three $\delta^{15}N$-$NH_4^+$ treatments. In fact, the observed $\delta^{15}N$ differences were not constant and the slope of $\delta^{15}N$-$NH_4^+$ versus $\delta^{15}N$-NO was significantly lower than unity under both oxic and hypoxic conditions (Fig. 4). This suggests that sources other than $NH_4^+$ oxidation contributed to the observed net NO production. Although NO can be produced by numerous microbial and abiotic processes (Medinets et al., 2015), we argue that the other major NO





source is mostly likely related to $NO_3^-$ consumption. This is based on the observation of high $NO_3^-$ concentrations in both oxic and hypoxic incubations, as well as the estimated low $R_{OrgN/NH4}$ (Table 2), which indicates a low availability of labile organic N – another potential substrate for NO production (Stange et al., 2013) – in this agricultural soil. Therefore, based on the assumption that $NH_4^+$ oxidation and $NO_3^-$ consumption were the two

primary NO sources during the oxic and hypoxic incubations, a two-source isotope mixing model was used to relate the measured $\delta^{15}N$-NO values to the concurrently measured $\delta^{15}N$-$NH_4^+$ and $\delta^{15}N$-$NO_3^-$ values:

$$\delta^{15}N\text{-NO} = f_{NH4}\times(\delta^{15}N\text{-}NH_4^+ - {}^{15}\eta_{NH4/NO}) + (1 - f_{NH4})\times(\delta^{15}N\text{-}NO_3^- - {}^{15}\eta_{NO3/NO}) \qquad \text{Equation (5)}$$

where ${}^{15}\eta_{NH4/NO}$ and ${}^{15}\eta_{NO3/NO}$ are the net isotope effects for NO production from $NH_4^+$ oxidation and $NO_3^-$ consumption, respectively. Rearranging Equation (5) yields Equation (6):

$$\delta^{15}N\text{-NO} = f_{NH4}\times\delta^{15}N\text{-}NH_4^+ + (1 - f_{NH4})\times\delta^{15}N\text{-}NO_3^- - [f_{NH4}\times{}^{15}\eta_{NH4/NO} + (1 - f_{NH4})\times{}^{15}\eta_{NO3/NO}] \quad \text{Equation (6)}$$

$$ {}^{15}\eta_{comb} = f_{NH4}\times{}^{15}\eta_{NH4/NO} + (1 - f_{NH4})\times{}^{15}\eta_{NO3/NO} \qquad \text{Equation (7)}$$

$$\delta^{15}N\text{-NO} = f_{NH4}\times\delta^{15}N\text{-}NH_4^+ + (1 - f_{NH4})\times\delta^{15}N\text{-}NO_3^- - {}^{15}\eta_{comb} \qquad \text{Equation (8)}$$

Equation (6) essentially dictates that the $\delta^{15}N$-NO values can be modeled from the $\delta^{15}N$-$NH_4^+$ and $\delta^{15}N$-$NO_3^-$ values using a hypothetical isotope effect for NO production from the combined soil $NH_4^+$ and $NO_3^-$ pool (${}^{15}\eta_{comb}$; the last

term in Equation (6)) that is a mixing of ${}^{15}\eta_{NH4/NO}$ and ${}^{15}\eta_{NO3/NO}$ controlled by $f_{NH4}$ (Equation 7). Thus, assuming $f_{NH4}$ and ${}^{15}\eta_{comb}$ were constant in each incubation experiment, $f_{NH4}$ and ${}^{15}\eta_{comb}$ can be solved using the measured $\delta^{15}N$-NO, $\delta^{15}N$-$NH_4^+$, and $\delta^{15}N$-$NO_3^-$ values from all three $\delta^{15}N$-$NH_4^+$ treatments (Equation 8). $f_{NH4}$ was estimated to be 0.72 under the oxic incubation (Table 2), indicating that 72% of the measured net NO production was sourced from $NH_4^+$ oxidation, with the remainder being ascribed to $NO_3^-$ consumption. Under the hypoxic condition, the share of $NH_4^+$

oxidation decreased to 58% (Table 2). ${}^{15}\eta_{comb}$ was estimated to be 56‰ under the oxic condition and 51‰ under the hypoxic condition (Table 2). Combining the $\delta^{15}N$-based NO source partitioning with the estimated $R_{NH4/NO3}$ and $R_{NO3comp}$, we further estimated NO yield in $NH_4^+$ oxidation and $NO_3^-$ consumption, respectively, and where the results are illustrated according to the classic "hole-in-the-pipe" (HIP) concept (Fig 9) (Davidson and Verchot, 2000). NO yield was 1.3% in $NH_4^+$ oxidation and 3.2% in $NO_3^-$ consumption in the oxic incubation (Fig. 9; Table

2). Under the hypoxic condition, NO yield was increased to 5.2% in $NH_4^+$ oxidation and 6.1% in $NO_3^-$ consumption (Fig. 9; Table 2).

Most previous laboratory and field studies suggest that soil NO emissions are predominately driven by nitrification, whereas NO produced from denitrification is further reduced to $N_2O$ before it escapes to the soil surface (Kester et al., 1997; Skiba et al., 1997). The minor role of denitrification is largely deduced from the supposition that denitrification is activated only under wet soil conditions (Davidson and Verchot, 2000). However,

based on our $\delta^{15}N$-based NO source partitioning, about 30% of the net NO production was contributed by $NO_3^-$ consumption under oxic condition, highlighting the potential importance of denitrification in driving soil NO emissions under conditions not typically conducive to its occurrence. There is growing evidence that extensive anoxic microsites can develop in otherwise well-aerated soils due to micro-scale variability of $O_2$ demand and soil

texture-dependent gas diffusion limitations (Keiluweit et al. 2018). Although we would not predict high rates of heterotrophic respiration in this agricultural soil with low organic carbon, it is possible that rapid $O_2$ consumption by nitrification may outpace $O_2$ supply through diffusion in soil microsites, fostering development of anoxic niches in



close association with nitrification hot spots (Kremen et al., 2005). Based on $^{15}N$ labeling and direct $^{15}NO$ measurements using a gas chromatograph-quadrupole mass spectrometer, Russow et al. (2009) demonstrated that nitrification contributed about 70% of net NO production in a well-aerated, $NH_4^+$-fertilized silt loam, in strong agreement with our results based on natural abundance $\delta^{15}N$ measurements. An even lower contribution to NO production, e.g., 26 to 44%, has been reported for nitrification in organic, N-enrich forest soils incubated under oxic conditions (Stange et al., 2013). The persistence of denitrifying microsites in the studied soil is further corroborated by the nearly doubled net NO production from $NO_3^-$ consumption in the hypoxic incubation (Fig. 9). Importantly, the actual NO yield in denitrification might be much higher than those estimated for gross $NO_3^-$ consumption during the oxic and hypoxic incubations (i.e., 3.2% and 6.1%), as denitrification occurring in anoxic niches might only comprise a small fraction of the estimated $R_{NO3comp}$.

Interestingly, while $R_{NH4/NO3}$ was significantly lower in the hypoxic incubation, the net NO production from $NH_4^+$ oxidation was similar between the two incubation experiments, indicating a higher NO yield in nitrification when $O_2$ availability became limited (Fig. 9). However, mechanisms underlying the differential NO yield in nitrification are difficult to elucidate owing to the high complexity of biochemical pathways of NO production by AOB and AOA. In AOB, the prevailing view of $NH_3$ oxidation is that it occurs via a two-step enzymatic process, involving hydroxylamine ($NH_2OH$) as an obligatory intermediate (Fig. 10). The first step is catalyzed by $NH_3$ monooxygenase (AMO), which uses copper and $O_2$ to hydroxylate $NH_3$ to $NH_2OH$. Next, a multiheme enzyme, $NH_2OH$ oxidoreductase (HAO), catalyzes the four-electron oxidation of $NH_2OH$ to $NO_2^-$ via enzyme-bound nitroxyl ([HNO-Fe]) and nitrosyl ([NO-Fe]) intermediates (Lehnert et al., 2018) (Fig. 10). Under this 'NH$_2$OH obligate intermediate' model, NO emission was proposed to result from dissociation of NO from the enzyme-bound nitrosyl complex under high $NH_3$ and/or low $O_2$ conditions (Fig. 10) (Hooper et al., 2005; Beeckman et al., 2018). However, there is recent strong evidence that HAO generally catalyzes the three-electron oxidation of $NH_2OH$ to NO under both aerobic and anaerobic conditions; the HAO-produced NO is further oxidized to $NO_2^-$ by an unknown enzyme (Caranto et al., 2017). In this way, NO would not be a byproduct of incomplete $NH_2OH$ oxidation, but rather required as an obligatory intermediate for $NO_2^-$ production (Fig. 10). It was further proposed that AOB-encoded copper-containing NIR may catalyze the final one-electron oxidation of NO to $NO_2^-$ by operating in reverse (Lancaster et al., 2018). Under this 'NH$_2$OH/NO obligate intermediate' model, high intracellular NO concentrations arise when the rate of NO production outpaces the rate of its oxidation to $NO_2^-$, leading to NO leakage from cells. Consequently, under $O_2$ stress, decreases in the rate of NO oxidation to $NO_2^-$ might be expected, and this may explain the observed increase in nitrification NO yield in the hypoxic incubation. Additionally, some AOB strains can produce NO in a process termed 'nitrifier-denitrification', in which NO is produced through NIR-catalyzed $NO_2^-$ reduction and can be further reduced to $N_2O$ by AOB-encoded NOR (Wrage-Mönning et al., 2018) (Fig. 10). Compared to AOB, the $NH_3$ oxidation pathway in AOA remains unclear (Beeckman et al., 2018). The current model is that $NH_3$ is first oxidized by an archaeal AMO to $NH_2OH$ and subsequently converted to $NO_2^-$ by an unknown HAO counterpart (Kozlowski et al., 2016). NO seems to be mandatory for archaeal $NH_2OH$ oxidation and has been proposed to act as a co-substrate for the $NO_2^-$ production (Kozlowski et al., 2016). Consequently, NO is usually produced and immediately consumed with tighter control in AOA than in AOB (Kozlowski et al., 2016).





To shed further light on the inner workings of net NO production from $NH_4^+$, we turn to constraining $^{15}\eta_{NH4/NO}$. Specifically, the inherent linkage between $^{15}\eta_{comb}$, $^{15}\eta_{NH4/NO}$, and $^{15}\eta_{NO3/NO}$ (Equation 7) allows one to probe the relative magnitude of $^{15}\eta_{NH4/NO}$ and $^{15}\eta_{NO3/NO}$ using the determined $^{15}\eta_{comb}$ and $f_{NH4}$. Given that $NO_2^-$ was absent in the soil and that NO reduction in denitrification was likely associated with a small isotope effect (i.e., $^{15}\eta_{NOR}$; see above), $^{15}\eta_{NO3/NO}$ in the oxic and hypoxic incubations should mainly reflect $^{15}\eta_{NAR}$. Thus, by assigning

the entire possible range of the best-fit $^{15}\eta_{NAR}$ derived in the anoxic incubation (5 to 45‰; Fig. 7a) to $^{15}\eta_{NO3/NO}$, $^{15}\eta_{NH4/NO}$ was estimated to range from 60 to 76‰ in the oxic incubation and from 55 to 84‰ in the hypoxic incubation (Fig. 11). If we take one step further by assuming that both $^{15}\eta_{NO3/NO}$ and $^{15}\eta_{NH4/NO}$ were identical between the oxic and hypoxic incubations, then $^{15}\eta_{NO3/NO}$ and $^{15}\eta_{NH4/NO}$ could be uniquely determined to be 30‰ and 66‰, respectively (Fig. 11; Table 2). Thus, the relative magnitude of $^{15}\eta_{NO3/NO}$ and $^{15}\eta_{NH4/NO}$ provides insights into the

differential relationship between $\delta^{15}N\text{-}NH_4^+$ and $\delta^{15}N\text{-}NO$ across the three $\delta^{15}N\text{-}NH_4^+$ treatments in the oxic and hypoxic incubations (Fig. 4). In the oxic incubation, if we assume that $^{15}\eta_{NH4/NO} = 66‰$ and $^{15}\eta_{NO3/NO} = 30‰$, the $\delta^{15}N$ of NO produced from $NH_4^+$ oxidation under the low $\delta^{15}N\text{-}NH_4^+$ treatment (about -60‰) would be much lower than the $\delta^{15}N$ of NO from $NO_3^-$ consumption (about -38‰). However, under the high $\delta^{15}N\text{-}NH_4^+$ treatment, the $\delta^{15}N$ of $NH_4^+$-produced NO would increase to about -14‰ and be higher than $\delta^{15}N$ values of $NO_3^-$-produced NO (about -

26‰). Consequently, the production of NO from $NO_3^-$ consumption would "dilute" the $\delta^{15}N$ of total net NO production, pulling it to fall below the 1:1 line between the $\delta^{15}N\text{-}NH_4^+$ and $\delta^{15}N\text{-}NO$ values in Fig. 4. This "dilution effect" was more pronounced in the hypoxic incubation due to the lower $f_{NH4}$ (i.e., higher contribution of $NO_3^-$-produced NO) (Fig. 4).

        Therefore, under either oxic or hypoxic condition, the net NO production from $NH_4^+$ oxidation proceeded

with a large $^{15}\eta_{NH4/NO}$. As $NH_3$ oxidation to $NH_2OH$ was likely the rate-limiting step for the entire nitrification process, a fraction of the inferred large $^{15}\eta_{NH4/NO}$ can be accounted for by the isotope effect for $NH_3$ oxidation to $NH_2OH$, which should be similar to the estimated $^{15}\eta_{NH4/NO3}$ (e.g., 23 to 28‰). The residual isotope effect, on the order of 40‰, must therefore stem from additional bond forming/breaking during net NO production in $NH_3$ oxidation. This additional N isotope effect could be explained by $NO_2^-$ reduction catalyzed by AOB-encoded NIR if

NO was dominantly produced through the nitrifier-denitrification pathway (Fig. 10). However, provided that the two oxidation steps of nitrification were tightly coupled under both oxic and hypoxic conditions, it is unlikely that $NO_2^-$ would accumulate to high enough intracellular concentrations to trigger nitrifier-denitrification (Wrage-Mönning et al., 2018). Similarly, we would not expect any substantial fractionations to result from accumulation of intracellular $NH_2OH$ or enzyme-bound intermediate species (e.g., [HNO-Fe] and [NO-Fe]). Thus, we are left with either a large

and normal isotope effect for NO dissociation from its enzyme-bound precursor if NO production was mainly routed through the '$NH_2OH$ obligate intermediate' pathway or an inverse isotope effect associated with NO oxidation if NO itself was an obligatory intermediate required for $NO_2^-$ production (Fig. 10). With respect to the first possibility, if NO dissociation from the Fe active site of HAO is mainly controlled by an equilibrium reaction between NO and enzyme-bound nitrosyl species, the forward and backward reactions may occur with distinctively different isotope

effects, giving rise to an equilibrium isotope effect that favors partitioning of $^{14}N$ to the dissociated NO. However, expression of this equilibrium isotope effect would be largely suppressed by limited isotope exchange between the





two N pools due to the presumably transient presence of nitrosyl intermediate. Therefore, a partial expression of a large equilibrium isotope effect (e.g., > 40‰) would be required to explain the residual N isotopic fractionation during NO production in $NH_3$ oxidation. Alternatively, in regards to the second possibility, if we assume that the

enzyme-catalyzed oxidation of NO to $NO_2^-$ proceeds via an enzyme-bound transition state and that the transition state contains the newly formed N-O bond, an inverse isotope effect may result from more strongly bonded N atom in the transition state, for which there is precedent in the literature (i.e., $NO_2^-$ oxidation to $NO_3^-$; see above) (Casciotti et al., 2009). Moreover, the small NO yield observed in the oxic and hypoxic incubations would indicate a large consumption of NO (i.e., 95 to 99%). With this high level of NO consumption, an inverse isotope effect on the

order of -13 to -9‰ would be sufficient to account for the residual isotope effect for net NO production from $NH_4^+$. This inferred isotope effect is of similar magnitude to that reported for NXR-catalyzed $NO_2^-$ oxidation (i.e., -13‰) (Casciotti et al., 2009). However, to unambiguously determine the mechanisms giving rise to the large $^{15}\eta_{NH4/NO}$, further biochemical analyses will be needed to clarify the enzymatic pathways responsible for NO production by AOB and AOA under relevant soil conditions. Nonetheless, the results presented here provide evidence that

production of NO with low $\delta^{15}N$ values may be a characteristic feature of nitrification in $NH_4^+$-fertilized agricultural soils under both oxic and hypoxic conditions.

**5 Implications for NO emission from agricultural soils**

In this study, the net production rates and $\delta^{15}N$ values of NO were measured under a range of controlled laboratory conditions. The results provide insights into how stable N and O isotopes can be effectively used to understand the

reaction mechanisms by which NO is produced and consumed in soils. While nitrification is the commonly cited source for NO emissions from agricultural soils, the measured net NO production rates in this study highlight the great potential of abiotic $NO_2^-$ reduction and denitrification in driving NO production and release from agricultural soils and thus should not be overlooked when attributing field soil NO emissions. Indeed, because NO is a direct product or free intermediate in these processes, abiotic $NO_2^-$ reduction and denitrification may inherently have an

larger NO yield – that is, a bigger "hole" for NO leaking in the HIP model (Davidson and Verchot, 2000). We conclude that the isotope-based measurement and modeling framework established in this work is a powerful tool to bridge NO production with gross N transformation processes in agricultural soils, thereby providing a quantitative way to parameterize the HIP model for modeling soil NO emissions under dynamic environmental conditions (e.g., varying temperature and soil moisture content).

The differences in the net isotope effects for NO production from abiotic $NO_2^-$ reduction, denitrification, and nitrification revealed in this study (Fig. 12a) suggest that $\delta^{15}N$-NO is a useful tracer for informing NO production pathways in agricultural soils. Specifically, the relatively small magnitude of $^{15}\eta_{NO2/NO(abiotic)}$ indicates that $\delta^{15}N$-NO is particularly useful in probing the relative importance of NO production from abiotic versus microbial reactions, lending support to our previous finding based on rewetting of a dry forest soil that high $\delta^{15}N$

values of rewetting-triggered NO pulses was mainly contributed by chemical $NO_2^-$ reduction (Yu and Elliott, 2017). Moreover, the large $^{15}\eta_{NH4/NO}$ revealed in the oxic and hypoxic incubations provides an empirical basis for discerning the relative role of $NH_4^+$ oxidation and $NO_3^-$ reduction in driving soil NO production and emissions.





Interestingly, comparing the measured net isotope effects for NO production from abiotic $NO_2^-$ reduction, denitrification, and nitrification with those previously quantified for $N_2O$ production in soil incubations and pure
cultures (Denk et al., 2017 and references therein; Jones et al., 2015; Wei et al., 2019), a similar pattern is evident across these three common production pathways for NO and $N_2O$ (Fig. 12a). This similarity reflects the intimate connection between NO and $N_2O$ turnovers within each reaction pathway and provides strong evidence that simultaneous $\delta^{15}N$-NO and $\delta^{15}N$-$N_2O$ measurements can potentially yield unprecedented insights into the sources and processes controlling NO and $N_2O$ emissions from agricultural soils. However, on the other hand, the
demonstrated reaction reversibility between $NO_2^-$ and $NO_3^-$ under anoxic conditions is a new complication that needs to be considered when using $\delta^{15}N$ to examine soil NO and $N_2O$ emissions. As $NO_2^-$ is often accumulated in agricultural soils following fertilizer application (Venterea et al., 2020), expression of the equilibrium isotope effect between $NO_2^-$ and $NO_3^-$ in redox-dynamic surface soils may render $\delta^{15}N$-NO and $\delta^{15}N$-$N_2O$ less useful in tracing NO and $N_2O$ sources. Given that high soil $NO_2^-$ concentrations can trigger emission pulses of NO and $N_2O$ (Venterea et
al., 2020), $NO_2^-$ accumulation should be taken as a critical sign for careful evaluation of the reaction complexity underlying $\delta^{15}N$ distributions among the denitrification products.

To further assess the potential utility of $\delta^{15}N$ measurements in source partitioning NO emissions from agricultural soils, we applied the estimated N isotope effects to the in situ $\delta^{15}N$-$NO_x$ measurements reported by Miller et al. (2018). Importantly, the soil used in this study was collected from the same farm where Miller et al.
(2018) conducted their field measurements (e.g., the USDA-managed corn-soybean field in central Pennsylvania, USA). Hence, the derived isotope effects may be particularly relevant to their reported $\delta^{15}N$-$NO_x$ values due to similar soil microbial community structures. Because $NO_2^-$ accumulation was not reported by Miller et al. (2018), we consider nitrification and denitrification to be the primary sources for the observed NO (and, to a much less extent, $NO_2$) emissions. Therefore, the $^{15}\eta_{NH4/NO}$ and $^{15}\eta_{NO3/NO}$ values derived in the oxic and hypoxic incubations
(i.e., 66‰ and 30‰, respectively) were used in combination with the $\delta^{15}N$ values of soil $NH_4^+$ and $NO_3^-$ reported in Miller et al. (2018) to calculate the $\delta^{15}N$ endmembers for NO produced from $NH_4^+$ oxidation and $NO_3^-$ reduction. As shown in Fig. 12b, comparing the in situ $\delta^{15}N$-$NO_x$ measurements with the estimated isotopic endmembers provides a compelling picture of soil NO dynamics following manure application. Notably, the initial low $\delta^{15}N$-$NO_x$ values reported by Miller et al. (2018) might indicate a mixed contribution of $NH_4^+$ oxidation and $NO_3^-$ reduction to soil
$NO_x$ emissions (Fig. 12b). Nevertheless, the increase in $\delta^{15}N$-$NO_x$ values measured 4 to 11 d after manure application may reflect a shift in dominant NO production pathway to denitrification, in line with the increasing accumulation of $NO_3^-$ supplied by nitrification in the soil (Miller et al., 2018). Although data-limited, this example provides promising initial evidence for the ability of multi-species $\delta^{15}N$ measurements to provide mechanistic information on soil NO dynamics and its environmental controls. Further experimental constraints on soil $\delta^{15}N$-NO
variations can build on the measurement and modeling framework developed in this study to advance our understanding of soil NO source contributions over a wide range of environmental conditions and soil types.

*Data availability*. The datasets generated for this study and documentation about the equations and parameters of the isotopologue-specific models are available in the Supplement.




*Supplement*. The supplement related to this article is available online at:

*Author contributions*. Z.Y. and E.M.E. designed the study; Z.Y. conducted the experiments and analyzed the data; Z.Y. and E.M.E. wrote the paper.


*Competing interests*. The authors declare no conflict of interest.

*Acknowledgements*. The authors thank Dr. Curtis Dell (USDA-ARS) for helping with the field soil sampling, and Katherine Redling, Vivian Feng, Madeline Ellgass, and Madeline Gray (University of Pittsburgh) for assistance with the isotopic analyses.


*Financial statement*. This work was supported by a National Science Foundation CAREER award (Grant No. 1253000) to E.M.E.

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





**Tables**

Table 1. Mean and 95% confidence interval of modeled denitrification rates and $NO_2^-$ re-oxidation rate constant under the 'no exchange' and 'complete exchange' scenarios.

| Parameter | Description | No exchange | | Complete exchange | |
|---|---|---|---|---|---|
| | | Mean | 95% CI | Mean | 95% CI |
| $R_{NAR}$ | Zero order rate for $NO_3^-$ reduction ($\mu g \cdot N \cdot g^{-1} \cdot h^{-1}$) | 0.158 | 0.157 to 0.160 | 0.158 | 0.157 to 0.160 |
| $R_{NIR}$ | Zero order rate for $NO_2^-$ reduction ($\mu g \cdot N \cdot g^{-1} \cdot h^{-1}$) | 0.112 | 0.111 to 0.113 | 0.112 | 0.111 to 0.113 |
| $R_{NOR}$ | Zero order rate for NO reduction ($\mu g \cdot N \cdot g^{-1} \cdot h^{-1}$) | 0.039 | 0.038 to 0.040 | 0.039 | 0.038 to 0.040 |
| $k_{NXR(b)}$ | First order rate constant of $NO_2^-$ re-oxidation ($h^{-1}$) | 0.64 | 0.61 to 0.66 | 0.25 | 0.24 to 0.26 |





Table 2. Mean and 95% confidence interval of modeled gross N transformation rates, NO yield, and net N isotope

effects in the oxic and hypoxic incubations.

| Parameter | Description | Oxic | | Hypoxic | |
|---|---|---|---|---|---|
| | | Mean | 95% CI | Mean | 95% CI |
| $R_{OrgN/NH4}$ | Zero order rate for net mineralization ($\mu g\ N\cdot g^{-1}\cdot h^{-1}$) | 0.014 | 0.013 to 0.016 | 0.012 | -0.011 to 0.038 |
| $R_{NH4/NO3}$ | Zero order rate for gross nitrification ($\mu g\ N\cdot g^{-1}\cdot h^{-1}$) | 0.458 | 0.455 to 0.460 | 0.111 | 0.110 to 0.113 |
| $R_{NO3comp}$ | Zero order rate for gross $NO_3^-$ consumption ($\mu g\ N\cdot g^{-1}\cdot h^{-1}$) | 0.071 | 0.070 to 0.072 | 0.070 | 0.049 to 0.091 |
| $^{15}\eta_{OrgN/NH4}$ | Net N isotope effect for net mineralization | 2‰ | -27 to 31‰ | 0‰ | -18 to 17‰ |
| $^{15}\eta_{NH4/NO3}$ | Net N isotope effect for gross nitrification | 28‰ | 27 to 30‰ | 23‰ | 12 to 33‰ |
| $^{15}\eta_{NO3comp}$ | Net N isotope effect for gross $NO_3^-$ consumption | 5‰ | -16 to 20‰ | 7‰ | -9 to 23‰ |
| $f_{NH4}$ | Fraction of net NO production from nitrification | 0.72 | 0.65 to 0.78 | 0.58 | 0.55 to 0.61 |
| $Y_{NH4/NO}$ | NO yield in nitrification | 1.3% | 1.2 to 1.4% | 5.2% | 4.8 to 5.5% |
| $Y_{NO3/NO}$ | NO yield in $NO_3^-$ consumption | 3.2% | 2.5 to 4.0% | 6.1% | 4.3 to 9.3% |
| $^{15}\eta_{comb}$ | Combined net isotope effect for NO production from $NH_4^+$ and $NO_3^-$ | 56‰ | 54 to 58‰ | 51‰ | 50 to 52‰ |
| | | Mean | | 95% CI | |
| $^{15}\eta_{NH4/NO}$ | Net isotope effect for NO production from $NH_4^+$ oxidation | 66‰ | | 59 to 85‰ | |
| $^{15}\eta_{NO3/NO}$ | Net isotope effect for NO production from $NO_3^-$ consumption | 30‰ | | 1 to 42‰ | |



**Figures**

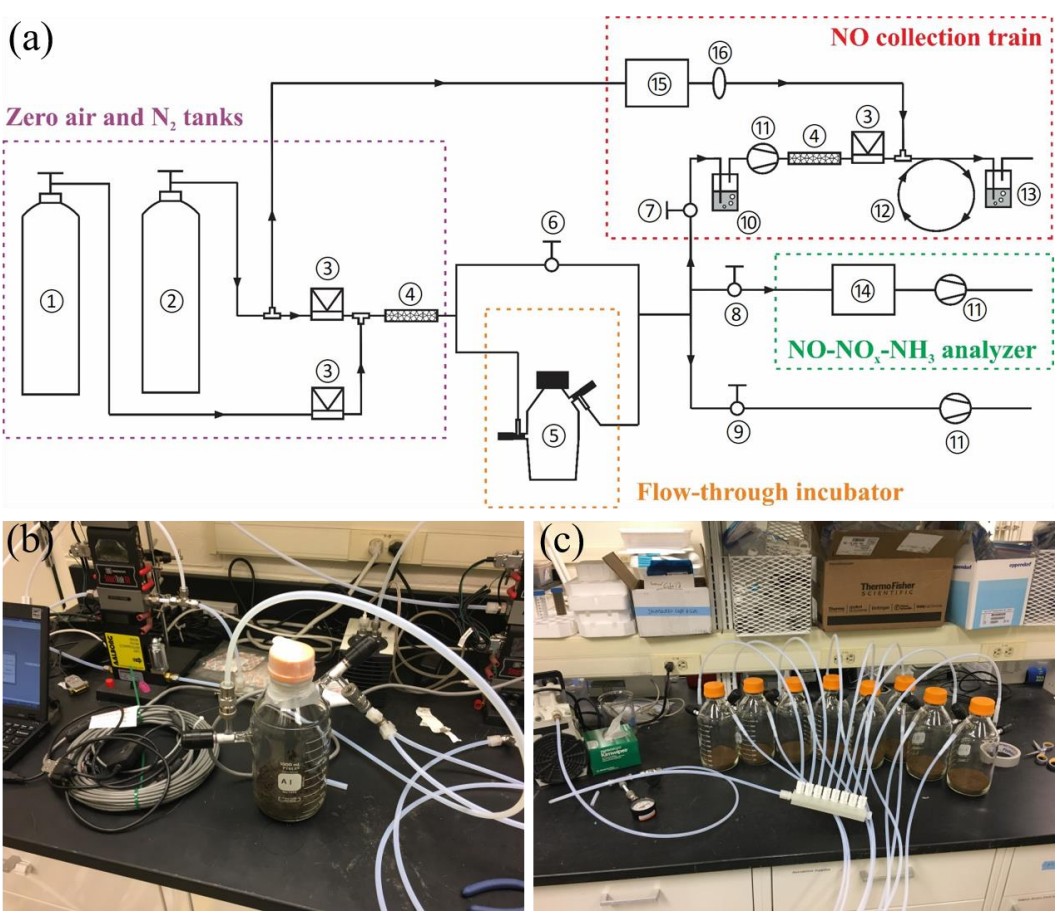

Figure 1. (a) Schematic of the DFC system (not to scale) consisting of the following: (1) zero air tank, (2) N2 tank,
(3) mass flow controller, (4) Nafion moisture exchanger, (5) flow-through incubator, (6) to (9) needle valves for
controlling vacuum and flushing of the DFC system, (10) HONO scrubber, (11) diaphragm pump, (12) Teflon
reaction tube, (13) gas washing bottle containing TEA solution, (14) NO-NO$_x$-NH$_3$ analyzer, (15) O$_3$ generator, (16)
in-line PTFE particulate filter assembly. (b) Photo of the flow-through incubator. (c) Photo of the Teflon purging
manifold for connection of the incubators in parallel.

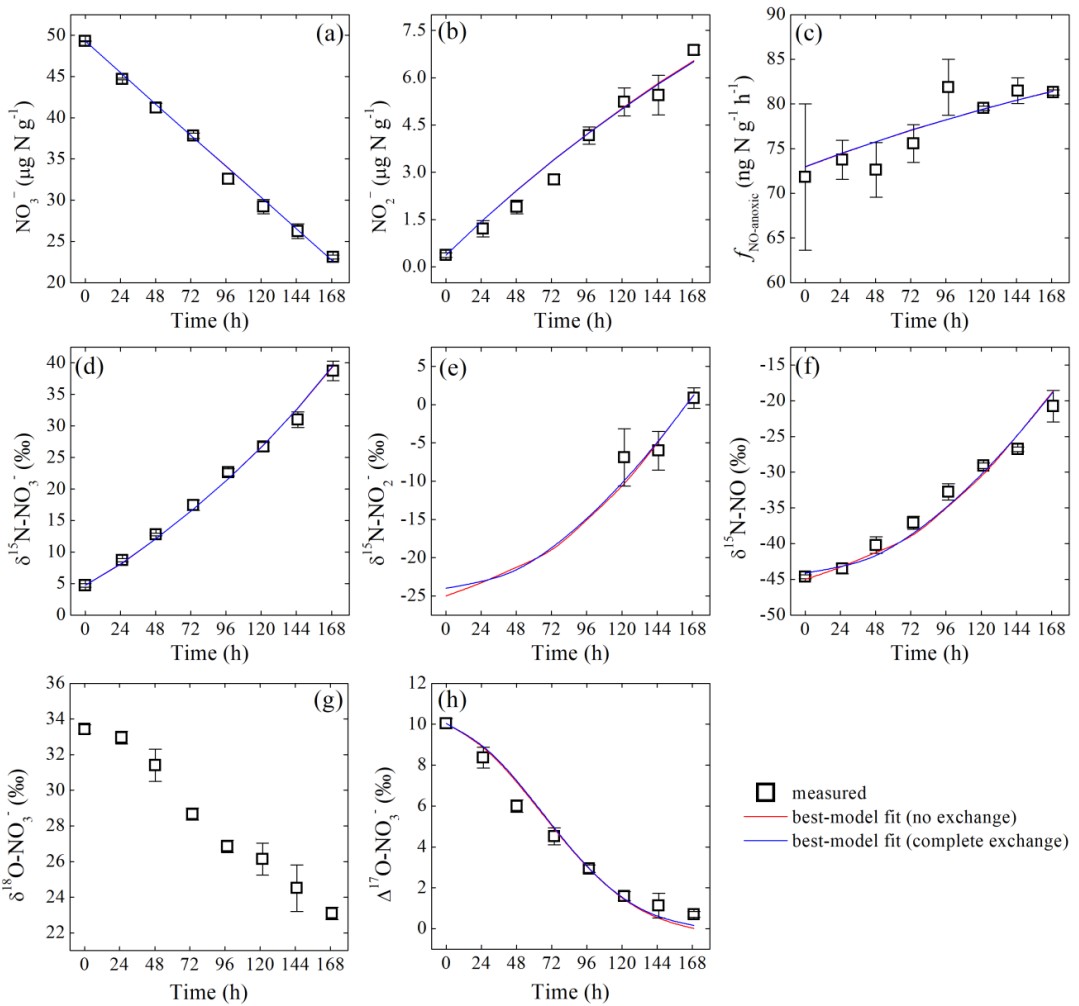

Figure 2. Measured and modeled concentrations of $NO_3^-$ (a) and $NO_2^-$ (b), net NO production rate (c),

$\delta^{15}N$ values of $NO_3^-$ (d), $NO_2^-$ (e), and NO (f), and $\delta^{18}O$ (g) and $\Delta^{17}O$ (h) of $NO_3^-$ during the anoxic

incubation.



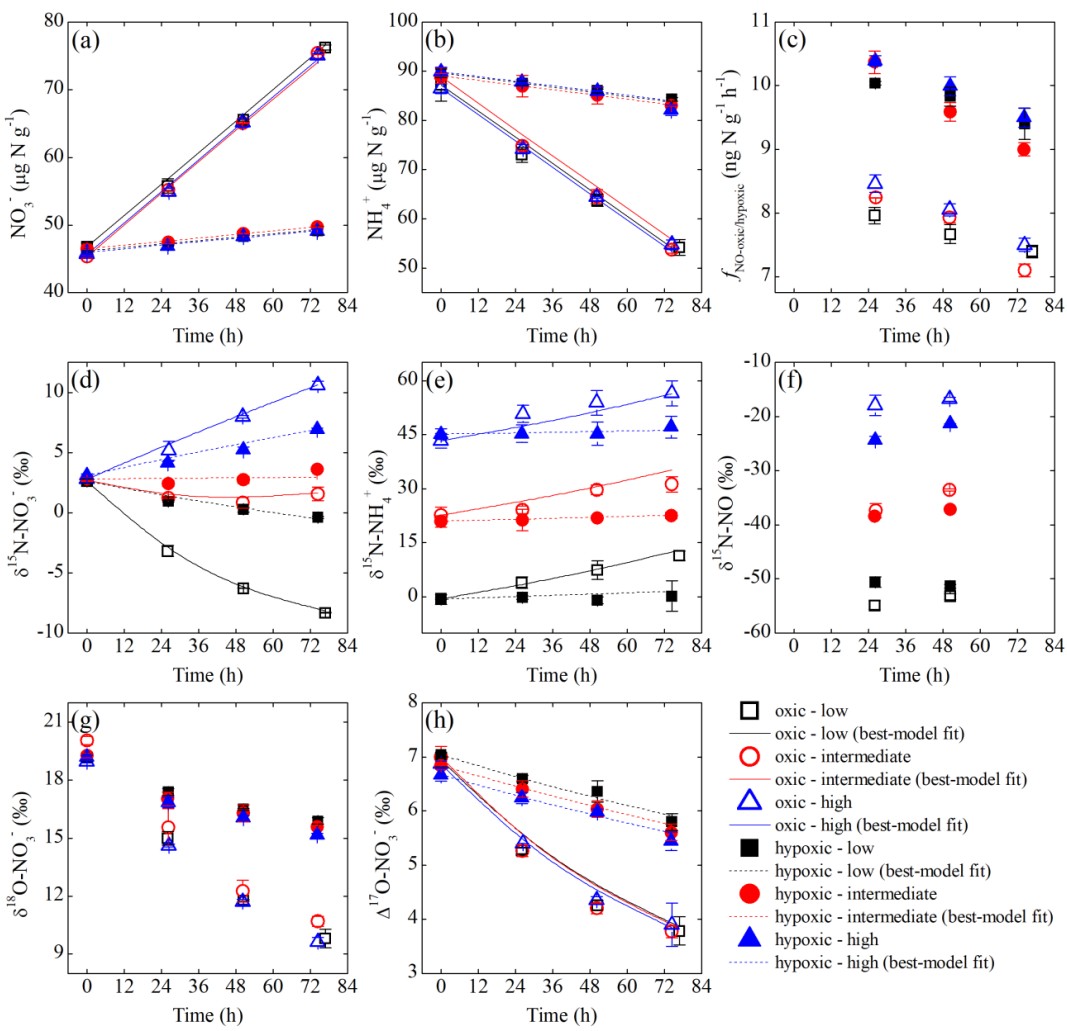

Figure 3. Measured and modeled concentrations of $NO_3^-$ (a) and $NH_4^+$ (b), net NO production rate (c),

$\delta^{15}N$ values of $NO_3^-$ (d) and $NH_4^+$ (e), and NO (f), and $\delta^{18}O$ (g) and $\Delta^{17}O$ (h) of $NO_3^-$ under the three $\delta^{15}N$-

$NH_4^+$ treatments (differed by color) of the oxic (open symbols) and hypoxic (solid symbols) incubations.





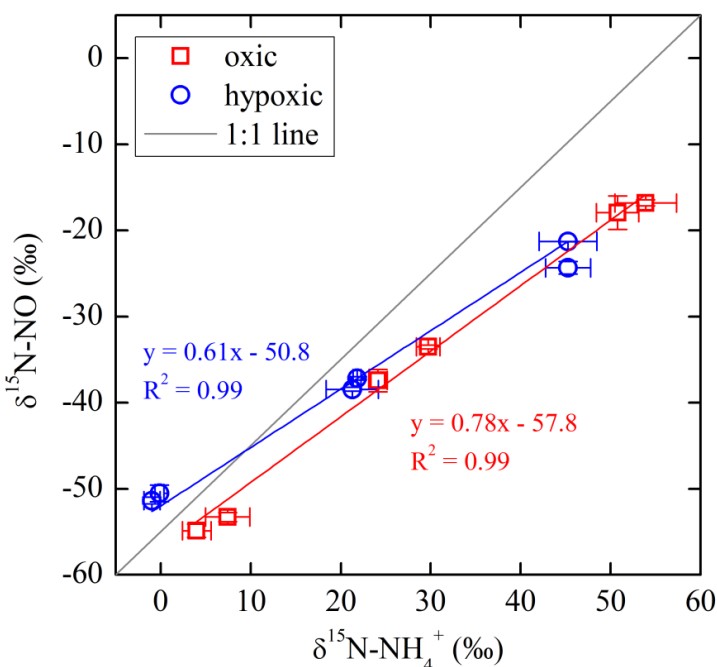

Figure 4. $\delta^{15}$N-NO as a function of $\delta^{15}$N-NH$_4^+$ in the oxic and hypoxic incubations.





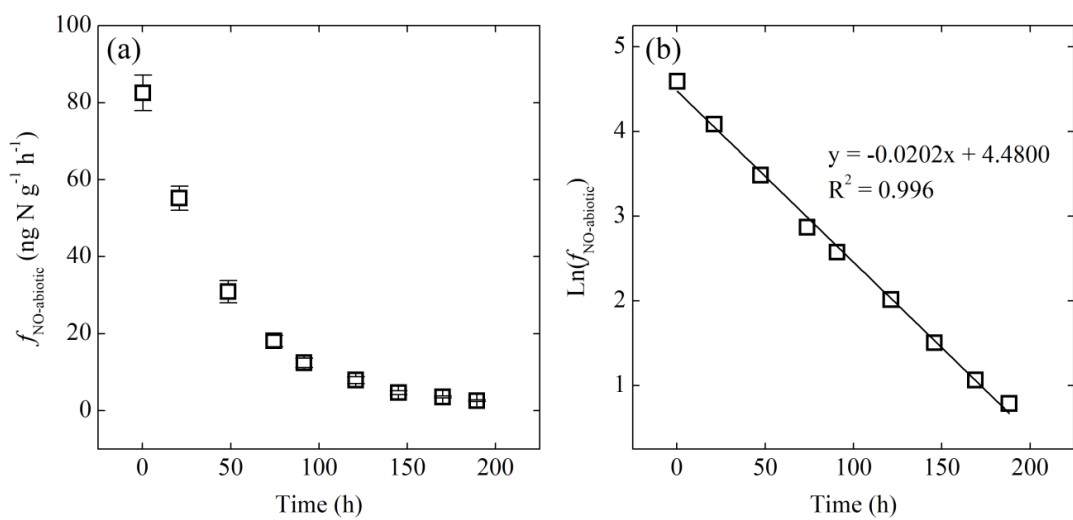

Figure 5 (a) Net NO production rate ($f_{NO\text{-}abiotic}$) of the $NO_2^-$-amended sterilized soil as a function of time.

(b) Plot of the natural logarithm of $f_{NO\text{-}abiotic}$ versus time showing first-order decay of $f_{NO\text{-}abiotic}$.





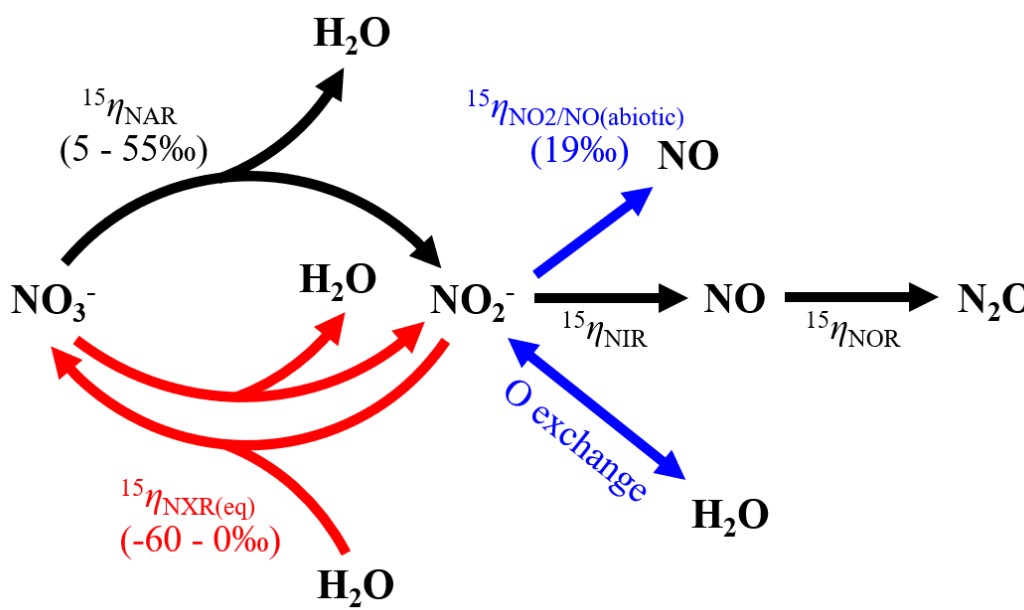

Figure 6. Model structure of co-occurring denitrification and $NO_2^-$ re-oxidation and associated N isotope

effects.



Figure 7. Contour maps showing variations in error-weighted residual sum of squares (RSS) between simulated and measured $\delta^{15}N$ values, modeled $^{15}\eta_{NIR}$, and modeled $^{15}\eta_{NOR}$ as a function of prescribed $^{15}\eta_{NAR}$ and $^{15}\eta_{NXR}$ under the 'no exchange' (a, c, and e) and 'complete exchange' (b, d, and f) model scenarios. Bold contour lines encompass the best-fit models defined by the lower 2.5th percentile of the error-weighted RSS.





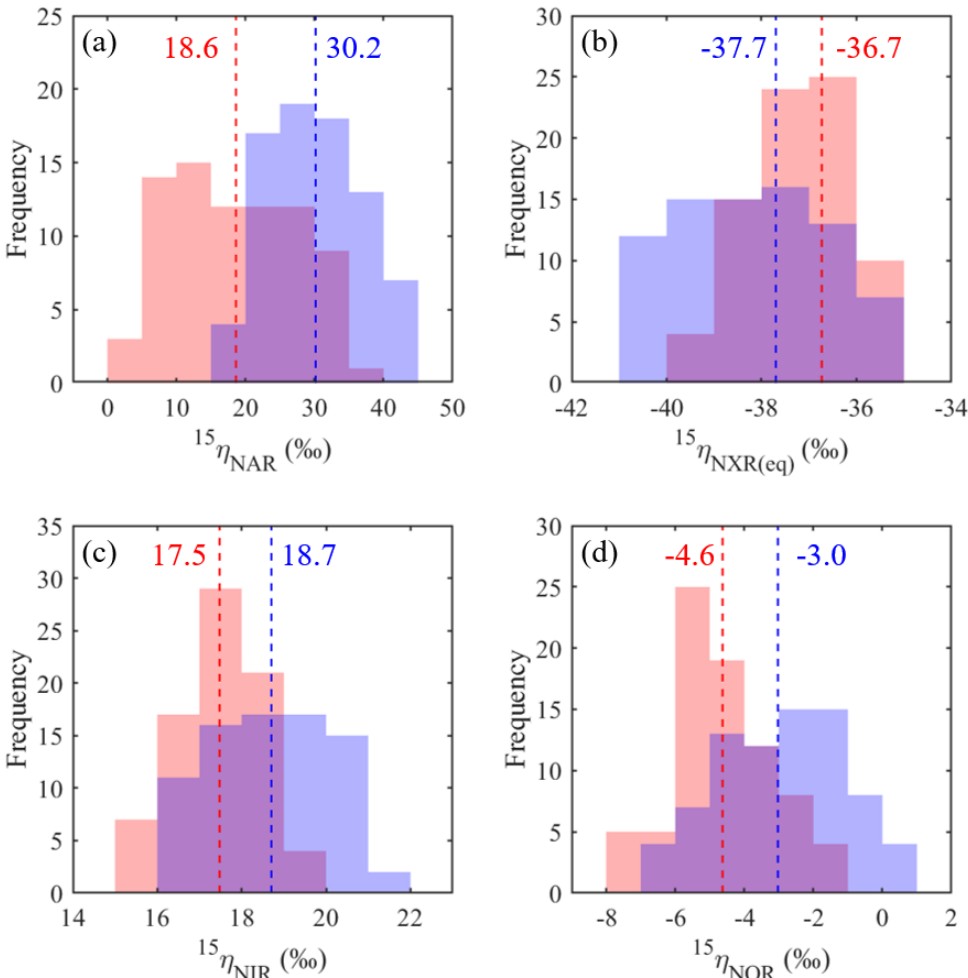

1142

Figure 8. Frequency distributions of the best-fit $^{15}\eta_{NAR}$ (a), $^{15}\eta_{NXR(eq)}$ (b), $^{15}\eta_{NIR}$ (c), and $^{15}\eta_{NOR}$ (d) under the 'no exchange' (red) and 'complete exchange' (blue) model scenarios. Dashed vertical lines denote the RSS-weighted mean $^{15}\eta$ values from the best-fit models under the two model scenarios.



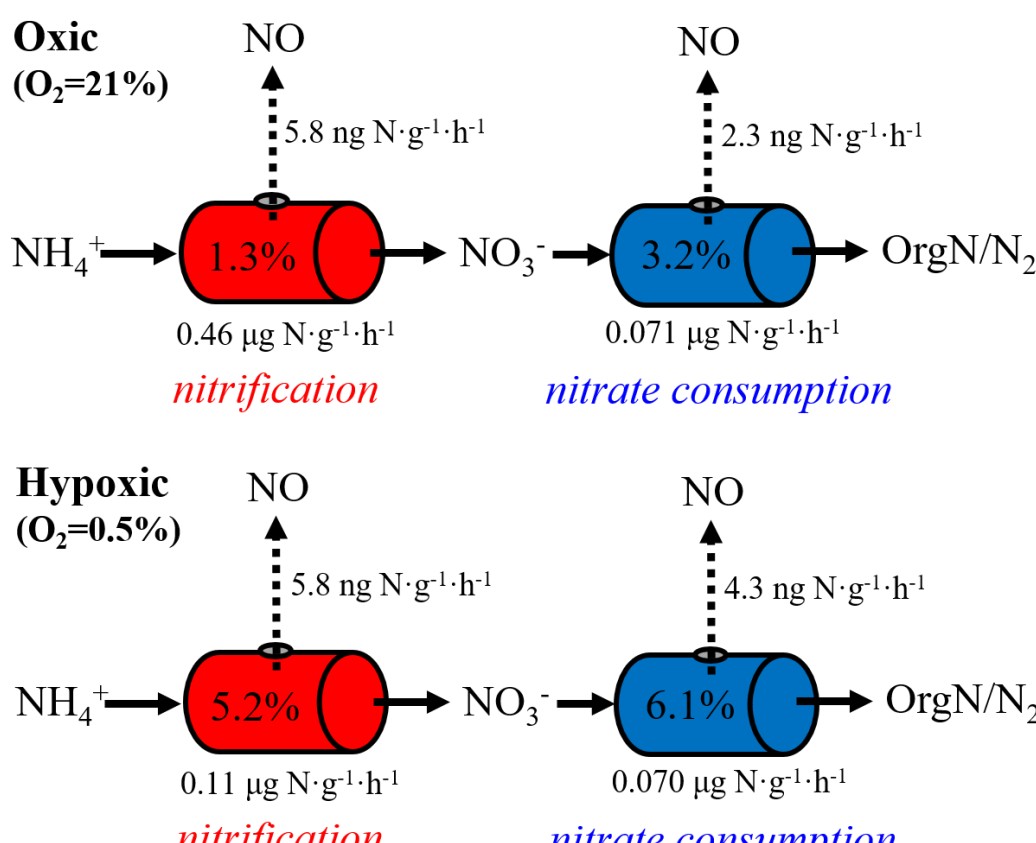

Figure 9. "Hole-in-the-pipe" illustration of NO production from gross nitrification and $NO_3^-$ consumption under oxic and hypoxic conditions. "OrgN" denotes organic nitrogen.



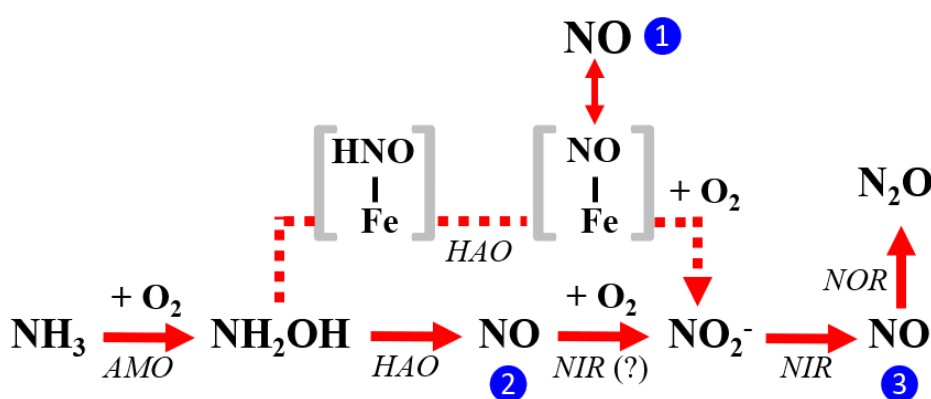

1149

Figure 10. Three enzymatic pathways for NO production during $NH_3$ oxidation to $NO_2^-$ by AOB: the 'NH$_2$OH obligatory intermediate' pathway indicated by blue circle (1), the 'NH$_2$OH/NO obligatory intermediate' pathway indicated by blue circle (2), and 'nitrifier-denitrification' pathway indicated by blue circle (3). Square brackets enclose proposed enzyme-bound intermediates [HNO-Fe] and [NO-Fe] of the 'NH$_2$OH obligatory intermediate' pathway. The role of AOB-encoded nitrite reductase (NIR) in catalyzing NO oxidation to $NO_2^-$ in the 'NH$_2$OH/NO obligatory intermediate' pathway is hypothetical.


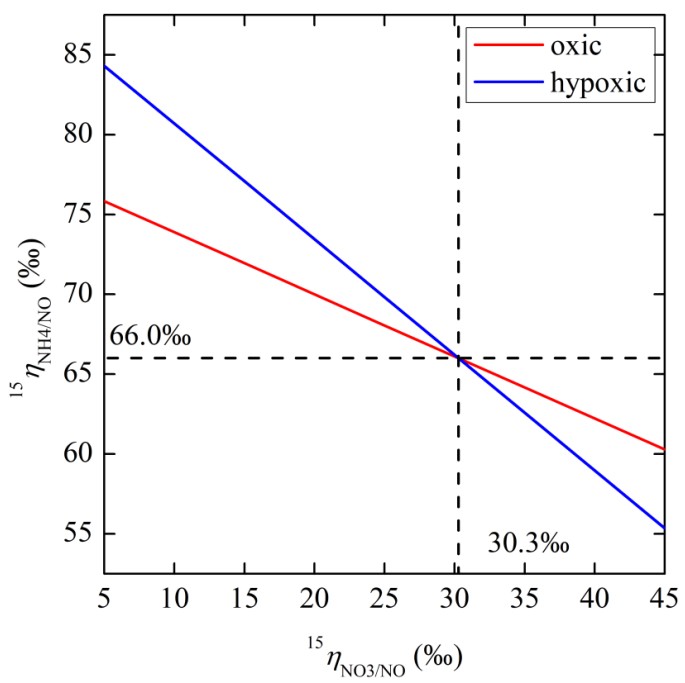

1156

Figure 11. Relative magnitude of net N isotope effects for NO production from $NH_4^+$ oxidation ($^{15}\eta_{NH4/NO}$)

and $NO_3^-$ consumption ($^{15}\eta_{NO3/NO}$) in the oxic and hypoxic incubations.



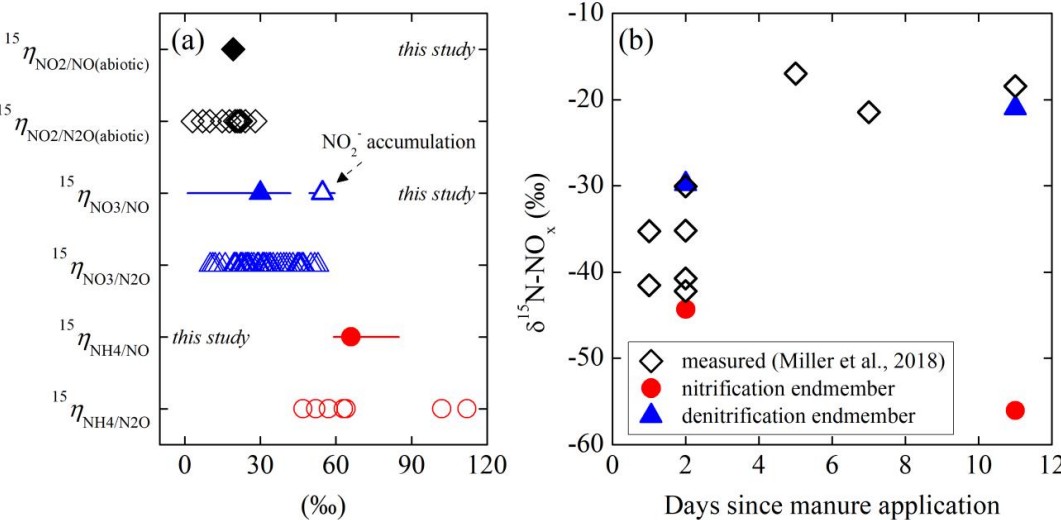

Figure 12. (a) Comparison of net isotope effects for NO production estimated in this study to net isotope effects for N₂O production reported in the literature. (b) Comparison of in situ $\delta^{15}N$ of $NO_x$ emission from a manure-fertilized soil (reported by Miller et al. (2018)) to nitrification and denitrification $\delta^{15}N$-NO endmembers derived using the estimated net isotope effects for NO production in the oxic and hypoxic incubations.