# Peer review of "Nitrogen isotopic fractionations during nitric oxide production in an agricultural soil"

_Biogeosciences, 2020_

## Referee Comment (RC1) · Anonymous Referee #1 · 18 Nov 2020

Dear authors, this was a pleasure to review your manuscript. It raises a very interesting topic of application of stable isotope studies for better understanding of soil N cycle. The manuscript presents a few of very original analytical approaches, like NO and NO2- isotopic analyses (as one of the very first for soil studies) and application of D17O to trace NO3 and NO2 soil transformations. The combination of all the approaches and the construction of the NO isotope model is very complex and challenging to present in an understandable form, but authors managed this very well. The manuscript is well organised, the results are well documented and supplement contains a lot of additional information precious for the readers who will further apply or develop the presented approach. I could have one suggestion of expanding the analytics, maybe useful for

your future studies. Since you used Chilian NO3 with the D17O anomaly you could also monitor this anomaly in NO2- (this may be difficult due to low concentrations) or in NO or N2O. This would allow you to determine the extend O-exchange and no further consideration of two scenarios: with and without O-exchange will be needed. This will bring more clarity to the whole study. An example of using D17O of N2O to determine O-exchange can be found in Lewicka-Szczebak et al. (2016, BG). I have just a few very minor comments: - Fig. 6 - do you assume that the abiotic NO cannot be further reduced to N2O? - L 609 - what do you mean here with "modified isotopologue-specific model" - this term was not used before in the manuscript and it is not clear if you just refer to the presented NO isotope model or sth else - L 624 - what is "more normal" isotope effect?  - Section 4.3 - I wonder why you do not consider NO2- oxidation to NO3- for oxic and suboxic conditions. If this process was so intensive under anoxic conditions, why it should not be active under oxic and suboxic conditions?

---

## Referee Comment (RC2) · Wolfgang Wanek (Referee) · 20 Nov 2020

Review of bg-2020-344 by Yu and Elliott "Nitrogen isotopic fractionations during nitric oxide production in an agricultural soil"

The paper reports on the isotopic fractionation of source and sink processes underlying soil NO emissions, NO emissions being important for atmospheric chemistry and as a soil N loss pathway. I am impressed by this study, proving in-depth analysis of isotopic constraints on formation and consumption pathways of soil NO, and partitioning the contribution of nitrifiers and denitrifiers as well as abiotic reactions. The approach taken with aerobic, suboxic and anoxic soil incubations combined with inorganic N

additions in live and sterile soils, N and O isotope measurements in inorganic soil N and NO, amended by isotope fractionation and flux modeling provides a most complete assessment of NO source and sink processes. This study therefore highlights that stable isotope measurements in inorganic soil N with those in NO and N2O can help in source attribution of these important atmospheric gases.

Minor corrections can be found in the annotated PDF.

Lines 59-61: There are also complete ammonia oxidizing Nitrospira, that catalyze the whole nitrification reaction sequence from ammonia to nitrate in one organism (comammox bacteria). Line 80 and throughout the MS: it should always be kinetic isotope fractionation and equilibrium isotope fractionation. Line 189: please provide xg (RCF) instead of rpm. Line 374: The reference Zhu-Baker et al. (2015) is missing in the reference list and should be Zhu-Barker.

Please also note the supplement to this comment:
https://bg.copernicus.org/preprints/bg-2020-344/bg-2020-344-RC2-supplement.pdf

**Supplement:**

[revised manuscript text omitted]

---

## Author Comment (AC1) · 9 Dec 2020

**Reply to Referee #1**

We are grateful for the constructive comments and helpful suggestions of Referee #1. Below are detailed responses to all the comments and corresponding explanations of the revisions made to the manuscript. Line numbers cited in the replies (highlighted) refer to the revised manuscript document with tracked changes displayed (attached with this reply). Please also read the comments to Referee #2 (Dr. Wolfgang Wanek) for other revisions of the manuscript.

Dear authors, this was a pleasure to review your manuscript. It raises a very interesting topic of application of stable isotope studies for better understanding of soil N cycle. The manuscript presents a few of very original analytical approaches, like NO and NO2- isotopic analyses (as one of the very first for soil studies) and application of D17O to trace NO3 and NO2 soil transformations. The combination of all the approaches and the construction of the NO isotope model is very complex and challenging to present in an understandable form, but authors managed this very well. The manuscript is well organised, the results are well documented and supplement contains a lot of additional information precious for the readers who will further apply or develop the presented approach.

**_Reply_:** We thank Referee #1 for the positive feedback.

**Comment 1:** I could have one suggestion of expanding the analytics, maybe useful for your future studies. Since you used Chilian NO3 with the D17O anomaly you could also monitor this anomaly in NO2- (this may be difficult due to low concentrations) or in NO or N2O. This would allow you to determine the extend O-exchange and no further consideration of two scenarios: with and without O-exchange will be needed. This will bring more clarity to the whole study. An example of using D17O of N2O to determine O-exchange can be found in Lewicka-Szczebak et al. (2016, BG).

**_Reply_:** We agree with the Referee that $\Delta^{17}O$ analysis of $NO_2^-$ could provide valuable insights into the degree of oxygen isotope exchange between $NO_2^-$ and $H_2O$ during the anoxic incubation, thereby offering more constraints and confidence to the isotopic modeling. However, we had concerns about the feasibility of $\Delta^{17}O$-$NO_2^-$ analysis in this case because $NO_2^-$ in water samples can undergo oxygen isotope exchange with $H_2O$ during sample processing, preservation, and storage (e.g. even for samples frozen under -20°C) (Casciotti et al., 2007). Therefore, measuring soil $NO_2^-$ for its $\Delta^{17}O$ values is not trivial, and will require comprehensive efforts to demonstrate its robustness throughout the sequence of soil extraction, extract processing, and sample storage. These efforts can be largely facilitated by development of $\Delta^{17}O$-$NO_2^-$ reference materials, which are currently lacking.

      Analysis and interpretation of $\Delta^{17}O$ of soil NO are confounded by the ozone oxidation of NO to $NO_2$ during the NO collection and the fact that $NO_2$ is collected in the triethanolamine (TEA) solution as both $NO_2^-$ and $NO_3^-$. Therefore, $\Delta^{17}O$ or $\delta^{18}O$ of $NO_2^-$/$NO_3^-$ collected from soil emitted-NO does not contain direct information about soil NO turnover. These technical aspects have been extensively discussed in our original method paper (Yu and Elliott, 2017).

      We have revised the manuscript to include Lewicka-Szczebak et al. (2016) and to note that our understanding of $NO_2^-$ oxygen isotope exchange and reaction reversibility can benefit from robust soil $\Delta^{17}O$-$NO_2^-$ determination and calibration in the future (Line 569-575).

I have just a few very minor comments:

**Comment 2:** - Fig. 6 - do you assume that the abiotic NO cannot be further reduced to N2O?

*Reply:* Due to lack of direct observational constraints, we did not assume any specific production or consumption pathways for NO yield from abiotic $NO_2^-$ reactions in the isotopologue-specific model. As such, the model simulates net NO production, rather than gross rates. Specifically, based on the results from the abiotic incubation, we assumed that the net abiotic NO production from $NO_2^-$ followed a pseudo-first order kinetics with respect to $NO_2^-$ with an apparent stoichiometric coefficient for net NO production from $NO_2^-$ of 0.52 (Line 510-513 of the original manuscript). This modeling parameterization implicitly accounts for parallel or competing abiotic NO production pathways in the soil, as well as potential NO consumption through abiotic reactions (e.g., chemo-denitrification of NO to $N_2O$; Line 365-380 of the original manuscript). In the revised manuscript, we have revised Fig 6 and its caption to clarify that the modeled abiotic NO production represents net NO yield, rather than gross NO production.

**Comment 3:** - L 609 - what do you mean here with "modified isotopologue-specific model" - this term was not used before in the manuscript and it is not clear if you just refer to the presented NO isotope model or sth else

*Reply:* It is mentioned in the original manuscript that the isotopologue-specific model we used to simulate co-occurring denitrification and $NO_2^-$ re-oxidation was modified from a model of co-occurring nitrification and $NO_3^-$ consumption we developed previously for well-aerated soils (Line 492-495 of the original manuscript). We have removed "modified" here to prevent any confusion.

**Comment 4:** - L 624 - what is "more normal" isotope effect?

*Reply:* In this study, we follow the convention to define kinetic isotope effect (Line 78-82 of the original manuscript). Under this definition, a normal kinetic isotope effect occurs when reaction rate constant of light isotopologues is higher than that of heavy isotopologues. Thus, normal kinetic isotope effects are expressed by positive eta ($\eta$) values in this study, in opposition to inverse kinetic isotope effects, which have negative $\eta$ values. Here, our estimated isotope effect for nitric oxide reduction ($^{15}\eta_{NOR}$) is between -8‰ and 2‰, higher than the previously reported $^{15}\eta_{NOR}$ for fungal nitric oxide reductase (i.e. -14‰). We have revised the manuscript to clarify that "more normal" is used here to describe our estimated $^{15}\eta_{NOR}$ being closer to zero (Line 631).

**Comment 5:** - Section 4.3 - I wonder why you do not consider NO2- oxidation to NO3- for oxic and suboxic conditions. If this process was so intensive under anoxic conditions, why it should not be active under oxic and suboxic conditions?

*Reply:* We did not explicitly consider aerobic $NO_2^-$ oxidation to $NO_3^-$ under oxic and hypoxic conditions because $NO_2^-$ concentration was below the detection limit in both incubations (Line 315-317 of the original manuscript), suggesting that the two steps of nitrification (i.e. $NH_4^+$ oxidation to $NO_2^-$ and $NO_2^-$ oxidation to $NO_3^-$) were tightly coupled under these conditions (Line 651-653 of the original manuscript). Therefore, in the isotopologue-specific model of co-occurring nitrification and $NO_3^-$ consumption, the two nitrification steps were lumped into a gross flux of $NH_4^+$ oxidation to $NO_3^-$ (Line 655-659 of the original manuscript; Text S5 in the Supplement) (Yu and Elliott, 2018). The excellent agreement between the modeled and

measured data (i.e., $NH_4^+$ and $NO_3^-$ concentrations and $\Delta^{17}O\text{-}NO_3^-$; Figure 3) under both oxic and hypoxic conditions confirms that this model configuration is appropriate.

The NXR-catalyzed anaerobic $NO_2^-$ re-oxidation and/or $NO_3^-/NO_2^-$ interconversion, which prevailed in the anoxic incubation, are considered not important in the oxic and hypoxic incubations. The results from the anoxic incubation, together with findings from previous studies (e.g. Wunderlich et al., 2013), suggest that $NO_2^-$ accumulation coupled with $O_2$ deprivation is the key trigger of anaerobic $NO_2^-$ re-oxidation by nitrite-oxidizing bacteria (NOB). This point has been emphasized in multiple places throughout the manuscript (Line 502-505, 598-604, and 839-846 of the original manuscript). The lack of $NO_2^-$ accumulation in the oxic and hypoxic incubations suggests that NOB mainly performed aerobic $NO_2^-$ oxidation to gain energy.

**References**

[revised manuscript text omitted]

---

## Author Comment (AC2) · 9 Dec 2020

**Reply to Dr. Wolfgang Wanek**

We are grateful for the constructive comments and helpful suggestions of Dr. Wanek. Below are detailed responses to all the comments and corresponding explanations of the revisions made to the manuscript. Line numbers cited in the replies (highlighted) refer to the revised manuscript document with tracked changes displayed (attached with this reply). Please refer to our replies to Referee #1 for other revisions of the manuscript.

The paper reports on the isotopic fractionation of source and sink processes underlying soil NO emissions, NO emissions being important for atmospheric chemistry and as a soil N loss pathway. I am impressed by this study, proving in-depth analysis of isotopic constraints on formation and consumption pathways of soil NO, and partitioning the contribution of nitrifiers and denitrifiers as well as abiotic reactions. The approach taken with aerobic, suboxic and anoxic soil incubations combined with inorganic N additions in live and sterile soils, N and O isotope measurements in inorganic soil N and NO, amended by isotope fractionation and flux modeling provides a most complete assessment of NO source and sink processes. This study therefore highlights that stable isotope measurements in inorganic soil N with those in NO and N2O can help in source attribution of these important atmospheric gases.

***Reply***: We are grateful for the encouraging remarks and positive feedback.

**Comment 1:** Minor corrections can be found in the annotated PDF.

***Reply***: We have incorporated all the corrections and edits into the revised manuscript. Thank you.

**Comment 2:** Lines 59-61: There are also complete ammonia oxidizing Nitrospira, that catalyze the whole nitrification reaction sequence from ammonia to nitrate in one organism (comammox bacteria).

***Reply***: We agree with Dr. Wanek that recent breakthrough in discovering completely nitrifying Nitrospira has broadened our understanding of microbial nitrification (Daims et al., 2015). However, to our best knowledge, studies on trace gas production (mainly as $N_2O$) by comammox bacteria are just starting (Kits et al., 2019), and whether and how free NO can be produced and released from complete nitrification remain unknown. There is also postulation that the revealed high affinity of comammox bacteria to ammonia may indicate a better adaptation of comamox bacteria to low-nitrogen environments (Kits et al., 2017; Kuyper, 2017). Therefore, for the sake of simplicity, we prefer not to include comammox bacteria in the discussion. Importantly, because $NO_2^-$ concentration was below the detection limit during the oxic and hypoxic incubations (Line 315-317 of the original manuscript), the two nitrification steps were lumped into a gross flux of $NH_4^+$ oxidation to $NO_3^-$ in our isotopologue-specific model (Line 655-659 of the original manuscript; Text S5 in the Supplement) (Yu and Elliott, 2018). Thus, our modeling scheme of nitrification is not in conceptual conflict with complete nitrification.

**Comment 3:** Line 80 and throughout the MS: it should always be kinetic isotope fractionation and equilibrium isotope fractionation.

***Reply:*** Agreed. We have revised the manuscript to adopt a consistent use of isotope terminology.

**Comment 4:** Line 189: please provide xg (RCF) instead of rpm.
***Reply:*** We have converted rpm (2000) to RCF (3400*g*) in the revised manuscript (Line 190).

**Comment 5:** Line 374: The reference Zhu-Baker et al. (2015) is missing in the reference list and should be Zhu-Barker.
***Reply:*** Thank you. We have corrected this mistake and double-checked the entire reference list to ensure its accuracy.

[revised manuscript text omitted]